# Kernel Implicit Variational Inference

**Jiaxin Shi**[*][†], **Shengyang Sun**[*][‡], **Jun Zhu**[†]

[†]Department of Computer Science & Technology, THU Lab for Brain and AI, Tsinghua University
[‡]Department of Computer Science, University of Toronto
`shijx15@mails.tsinghua.edu.cn, ssy@cs.toronto.edu, dcszj@tsinghua.edu.cn`

## Abstract

Recent progress in variational inference has paid much attention to the flexibility of variational posteriors. One promising direction is to use implicit distributions, i.e., distributions without tractable densities as the variational posterior. However, existing methods on implicit posteriors still face challenges of noisy estimation and computational infeasibility when applied to models with high-dimensional latent variables. In this paper, we present a new approach named *Kernel Implicit Variational Inference* that addresses these challenges. As far as we know, for the first time implicit variational inference is successfully applied to Bayesian neural networks, which shows promising results on both regression and classification tasks.

## 1 Introduction

Bayesian methods have been playing vital roles in machine learning by providing a principled approach for generative modeling, posterior inference and preventing over-fitting (Ghahramani, 2015). As it becomes a common practice to build deep models that have many parameters (LeCun et al., 2015), it is even more important to have a Bayesian formulation to capture the uncertainty in these models. For example, Bayesian Neural Networks (BNNs) (Neal, 2012; Blundell et al., 2015) have shown promise in reasoning about model confidence and learning with few labeled data. Another recent trend is to incorporate deep neural networks as a powerful function mapping between random variables in a Bayesian network, such as deep generative models like variational autoencoders (VAE) (Kingma & Welling, 2013).

Except a few simple examples, Bayesian inference is typically challenging, for which variational inference (VI) has been a standard workhorse to approximate the true posterior (Zhu et al., 2017). Traditional VI focuses on factorized variational posteriors to get analytical updates (known as Mean-field VI). While recent progress in this field drives VI into stochastic, differentiable and amortized (Hoffman et al., 2013; Paisley et al., 2012; Mnih & Gregor, 2014; Kingma & Welling, 2013), which does not rely on analytical updates anymore, factorized posteriors are still commonly used as the variational family. This greatly restricts the flexibility of the variational posterior, especially in high-dimensional spaces, which often leads to biased solutions as the true posterior is usually not factorized, thus not in the family. There have been some works that try to improve the flexibility of variational posteriors, borrowing ideas from invertible transformation of probability distributions (Rezende & Mohamed, 2015; Kingma et al., 2016). In their works, it is important for the transformation to be invertible to ensure that the transformed distribution has a tractable density.

Although utilizing invertible transformation is a promising direction to increase the expressiveness of the variational posterior, we argue that a more flexible variational family can be constructed by using general deterministic or stochastic transformations, which are not necessarily invertible. As a common result, the variational posterior we get in this way does not have a tractable density, despite that there is a way to sample from it. This kind of distribution is called implicit distributions, and for variational methods that use an implicit variational posterior (also known as variational programs (Ranganath et al., 2016) or wild variational approximations (Liu & Feng, 2016)), we refer to them as *Implicit Variational Inference* (implicit VI). Most of the existing implicit VI methods (Mescheder et al., 2017; Huszár, 2017; Tran et al., 2017) rely on a discriminator to produce estimates of the

---

[*]These authors contribute equally; J.Z is the corresponding author.

variational objective and its gradients. As pointed out by many of them, the estimates are often noisy and can lead to unstable training. Besides, discriminator-based approaches are computationally infeasible when applied to nontrivial BNNs.

In this paper we present an approach named *Kernel Implicit Variational Inference* (KIVI), which addresses the noisy estimation problem in previous works by providing a principled way of tuning the bias-variance tradeoff. Furthermore, KIVI does not rely on a discriminator and thus is computationally feasible for models with high-dimensional latent variables (e.g., BNNs). KIVI is applicable to both global and local latent variable models, which is demonstrated by experiments on BNNs and VAEs. As far as we know, this is the first time that implicit VI is successfully applied to BNNs, which shows promising results on both regression and classification tasks.

## 2 BACKGROUND

Consider a generative model $p(\mathbf{z}, \mathbf{x}) = p(\mathbf{z})p(\mathbf{x}|\mathbf{z})$, where $\mathbf{x}$ and $\mathbf{z}$ denote observed and latent variables, respectively. In VI, a variational distribution $q_\phi(\mathbf{z})$ in some parametric family is chosen to approximate the true posterior $p(\mathbf{z}|\mathbf{x})$ by optimizing the *evidence lower bound* (ELBO):

$$\mathcal{L}(\mathbf{x}; \phi) = \mathbb{E}_{q_\phi(\mathbf{z})}[\log p(\mathbf{x}|\mathbf{z})] - \mathrm{KL}(q_\phi(\mathbf{z})\|p(\mathbf{z})), \tag{1}$$

where KL denotes the Kullback-Leibler divergence $\mathrm{KL}(q\|p) = \mathbb{E}_q[\log\frac{q}{p}]$. This objective is a lower bound of the log-likelihood $\log p(\mathbf{x})$ since it can be written as $\mathcal{L}(\mathbf{x}; \phi) = \log p(\mathbf{x}) - \mathrm{KL}(q_\phi(\mathbf{z})\|p(\mathbf{z}|\mathbf{x}))$. The maximum of this objective is achieved when $q_\phi(\mathbf{z}) = p(\mathbf{z}|\mathbf{x})$. From Eq. (1), we can see that the challenge of using an implicit $q_\phi$ is that calculating $\mathrm{KL}(q_\phi(\mathbf{z})\|p(\mathbf{z}))$ requires evaluating the density of $q_\phi$, which is intractable for an implicit distribution.

Recently, inspired by the probabilistic interpretation of Generative Adversarial Networks (GAN) (Goodfellow et al., 2014; Mohamed & Lakshminarayanan, 2016), there have been some works that extend the GAN approach to the posterior inference of latent variable models (LVMs) (Mescheder et al., 2017; Huszár, 2017; Tran et al., 2017). These methods all use an implicit variational family and thus can be categorized into implicit VI methods. One of their key observations is that the density ratio $\frac{q_\phi(\mathbf{z})}{p(\mathbf{z})}$ can be estimated from samples of the two distributions by a probabilistic classifier called the discriminator. They first assign class labels ($y$) to $q$ and $p$: Let samples from $q_\phi(\mathbf{z})$ be of class $y = 1$, and samples from $p(\mathbf{z})$ be of class $y = 0$. Given an equal class prior, the density ratio at a given point can be calculated as $q_\phi(\mathbf{z})/p(\mathbf{z}) = p(\mathbf{z}|y = 1)/p(\mathbf{z}|y = 0) = p(y = 1|\mathbf{z})/p(y = 0|\mathbf{z})$, which is the ratio between the class probabilities given the data point. To estimate this, a discriminator $D$ is trained to classify between the two classes, with a logistic loss:

$$\max_D \mathbb{E}_{q_\phi(\mathbf{z})}[\log(D(\mathbf{z}))] + \mathbb{E}_{p(\mathbf{z})}[\log(1 - D(\mathbf{z}))], \tag{2}$$

where $D(\mathbf{z})$ outputs the probability of $\mathbf{z}$'s being from class $y = 1$. Given that $D$ is sufficiently flexible, the optimal solution of Eq. (2) is $D(\mathbf{z}) = q_\phi(\mathbf{z})/(q_\phi(\mathbf{z}) + p(\mathbf{z}))$. Therefore, the KL divergence term in the ELBO of Eq. (1) can be approximated as $\mathrm{KL}(q_\phi\|p) \approx \mathbb{E}_{q_\phi(\mathbf{z})}[\log D(\mathbf{z}) - \log(1 - D(\mathbf{z}))]$. This is called prior-contrastive forms of VI in Huszár (2017). Note that the ratio approximation does not change the gradients once the approximation is accurate, as we shall see later in Eq. (7). Though incorporating the discriminative power in a probabilistic model has shown great success in GANs, this method still suffers from challenging problems when applied to VI:

- **Noisy density ratio estimation (DRE)** In VI, the variational posterior gets updated in each iteration. As shown in Eq. (2), the discriminator should be trained to optimum after each update. However, in practice the inner loop for training the discriminator is often truncated to one or several iterations. At the beginning of the inference procedure, it is hard for the discriminator to catch up with the variational posterior. The noisy signal produced by the discriminator leads to noisy gradients and thus unstable training. Besides, even if the discriminator quickly achieves the optimum in a small number of iterations, there is still another issue. Notice that the training loss in Eq. (2) is with expectations. But in practice we are using samples from the two distributions to approximate it. When the support of the distributions is high-dimensional, given the limited number of samples we use, the variance of this estimate is considerable, i.e., the discriminator tends to overfit the samples. The phenomenon is that the discriminator arrives at a state where samples are easily distinguished and the probabilities given by the discriminator are near 0 or 1, which is commonly observed in experiments (Mescheder et al., 2017).

- **Computationally infeasible for high dimensional latent variables** As the density ratio is estimated by a discriminator, the samples from the two distributions of latent variables should be fed into it. However, the typically used neural network discriminator cannot afford very high-dimensional inputs (e.g., parameters in a moderate-size Bayesian neural network).

## 3 KERNEL IMPLICIT VARIATIONAL INFERENCE

To address the above challenges for implicit VI, we propose to replace the discriminator with a kernel method for DRE. The advantages of this method are that it has a closed-form solution and that it allows us to explicitly tradeoff between bias and variance by tuning a regularization coefficient.

### 3.1 ESTIMATING THE KL TERM

Specifically, let $\mathbf{z} \in \mathbb{R}^d$ be the latent variable, and the true density ratio is $r(\mathbf{z}) = q_\phi(\mathbf{z})/p(\mathbf{z})$. Consider modeling it with a function $\hat{r} \in \mathcal{H}$, where $\mathcal{H}$ is a *Reproducing Kernel Hilbert Space* (RKHS) induced by a positive definite kernel $k(\mathbf{z}, \mathbf{z}') : \mathbb{R}^d \times \mathbb{R}^d \to \mathbb{R}$. Similar to kernel ridge regression, we use an objective composed of a squared loss for regression plus a penalty for the complexity of the function. For the squared loss we choose the form used by the unconstrained Least Square Importance Fitting (uLSIF) (Kanamori et al., 2009):

$$\mathcal{J}(\hat{r}) = \frac{1}{2} \int (\hat{r}(\mathbf{z}) - r(\mathbf{z}))^2 p(\mathbf{z}) \, d\mathbf{z} = \frac{1}{2} \, \mathbb{E}_p \hat{r}(\mathbf{z})^2 - \mathbb{E}_q \hat{r}(\mathbf{z}) + C, \tag{3}$$

where $C$ is a constant, and approximate the expectation in $\mathcal{J}(\hat{r})$ by Monte Carlo estimates:

$$\hat{\mathcal{J}}(\hat{r}) = \frac{1}{2n_p} \sum_{i=1}^{n_p} \hat{r}(\mathbf{z}_i^p)^2 - \frac{1}{n_q} \sum_{j=1}^{n_q} \hat{r}(\mathbf{z}_j^q) + C, \quad \mathbf{z}_i^p \sim p(\mathbf{z}), \ \mathbf{z}_j^q \sim q_\phi(\mathbf{z}),$$

where $n_p$ and $n_q$ are the number of samples from $p$ and $q$, respectively. Note that in Eq. (3), the expectation of the squared loss is taken w.r.t. $p$ so that the resulting form can be estimated without evaluating the density of both distributions. For the penalty term, the complexity of $\hat{r}$ is measured by its RKHS norm ($\|\hat{r}\|_{\mathcal{H}}^2$). Putting them together, we get the final objective:

$$\min_{\hat{r} \in \mathcal{H}} \ \hat{\mathcal{J}}(\hat{r}) + \frac{\lambda}{2} \|\hat{r}\|_{\mathcal{H}}^2. \tag{4}$$

Here $\lambda$ is the regularization coefficient.

**Proposition 1.** *The optimal solution of Eq. (4) lies in the linear subspace spanned by the kernel functions centered at the samples ($\{\mathbf{z}_i^p\}_{i=1}^{n_p}, \{\mathbf{z}_j^q\}_{j=1}^{n_q}$), i.e., $\hat{r}$ has the form:*

$$\hat{r} = \sum_{i=1}^{n_p} \alpha_i k(\mathbf{z}_i^p, \cdot) + \sum_{j=1}^{n_q} \beta_j k(\mathbf{z}_j^q, \cdot). \tag{5}$$

*Proof.* This can be seen as the generalization of the representer theorem (Schölkopf et al., 2001) to the density ratio problem. So the proof follows the same procedure. See Appendix A. □

Substituting Eq. (5) into Eq. (4) and setting the derivatives w.r.t. $\boldsymbol{\alpha}$ and $\boldsymbol{\beta}$ to zeros, we get the optimal solution (a detailed derivation is given in Appendix B):

$$\boldsymbol{\beta} = \frac{1}{\lambda n_q} \mathbf{1}, \quad \boldsymbol{\alpha} = -\frac{1}{\lambda n_p n_q} \left( \frac{1}{n_p} \mathbf{K}_p + \lambda \mathbf{I} \right)^{-1} \mathbf{K}_{pq} \mathbf{1}, \tag{6}$$

where $(\mathbf{K}_p)_{i,j} = k(\mathbf{z}_i^p, \mathbf{z}_j^p)$ and $(\mathbf{K}_{pq})_{i,j} = k(\mathbf{z}_i^p, \mathbf{z}_j^q)$. We use the common RBF kernels $k(\mathbf{z}, \mathbf{z}') = \exp\left(-\|\mathbf{z} - \mathbf{z}'\|_2^2 / 2\sigma^2\right)$. $\sigma$ is the kernel bandwidth, which is determined by the commonly used median heuristic (the median of pairwise distances of the sample points).

Once we have the approximate density ratio function $\hat{r}$, a Monte Carlo estimate of $\text{KL}(q_\phi(\mathbf{z})\|p(\mathbf{z}))$ can be constructed by $\frac{1}{n_q} \sum_{i=1}^{n_q} \log \hat{r}(\mathbf{z}_i^q)$. Note that there is a constraint that the estimated density ratio should be non-negative. However, we do not involve it in the optimization objective in order to

get a closed-form solution, which indicates that some post-processing is needed to ensure this property. We solve the issue by clipping the estimated density ratio. The clipping values are searched from $\{10^{-8}, 10^{-16}, 10^{-32}\}$. In experiments we found that the algorithm is not sensitive to the clipping value. This is due to the accurate estimation guaranteed by the global optimum in the RKHS, which is a universal family when RBF kernels are used (Carmeli et al., 2010).

**The reverse ratio trick**  Another technique is essential to improve the estimation of the KL term, which we call the reverse ratio trick. The key observation is that the expectation in the squared loss $\mathcal{J}(\hat{r})$ in Eq. (3) is taken w.r.t. $p$, whereas the expectation in the KL term (KL$(q\|p)$) is taken w.r.t. $q$. Unless $p$ and $q$ match very well in where they put most probabilities, a small squared loss does not always mean a good KL estimate. The solution is by a simple trick. Instead of estimating $\frac{q}{p}$, we choose to estimate $r_{pq} = \frac{p}{q}$ and compute the KL term as $-\mathbb{E}_q \log \frac{p}{q}$. We denote the estimated reverse density ratio as $\hat{r}_{pq}$, then the corresponding KL estimate is $-\mathbb{E}_q \log \hat{r}_{pq}$. Note that in this way the squared loss changes to $\mathcal{J}(\hat{r}_{pq}) = \frac{1}{2}\int(\hat{r}_{pq}(\mathbf{z}) - r_{pq}(\mathbf{z}))^2 q_\phi(\mathbf{z})\,d\mathbf{z}$, whose expectation is taken w.r.t. the same probability measure ($q$) as the KL term's. As we shall see in experiments (Appendix F.1), the trick is essential to make the estimation sufficiently accurate for VI.

**Gradient computation**  We now consider how to estimate the gradients of the KL term w.r.t. variational parameters $\phi$. First it is easy to prove as in Huszár (2017) that

$$\nabla_\phi \text{KL}(q_\phi\|p) = -\nabla_\phi \mathbb{E}_{q_\phi} \log \frac{p}{q_\phi} = -\nabla_\phi \mathbb{E}_{q_\phi} \log \frac{p}{q}. \tag{7}$$

A detailed proof is in Appendix C. Eq. (7) indicates that the true gradients of the KL term w.r.t. $\phi$ do not flow through the density ratio function. We now replace the ratio on the right side with $\hat{r}_{pq}$: $\nabla_\phi \text{KL}(q_\phi\|p) \approx -\nabla_\phi \mathbb{E}_{q_\phi} \log \hat{r}_{pq}$. Note that without Eq. (7) we cannot do the approximation since $\hat{r}_{pq}$ has zero gradients w.r.t. $\phi$. Then, the reparameterization trick (Kingma & Welling, 2013) can be used:

$$-\nabla_\phi \mathbb{E}_{q_\phi} \log \hat{r}_{pq} = -\mathbb{E}_{\epsilon \sim N(0,I)} \nabla_\phi \log \hat{r}_{pq}(\mathbf{z}^q(\epsilon; \phi)).$$

## 3.2 THE ALGORITHM

We have constructed a closed-form estimate for the KL term and show its gradients can be estimated by the reparameterization trick. Note that the reparameterization trick can also be used to compute the gradients of the reconstruction term in Eq. (1) and thus can be applied to the ELBO. See Algo. 1 for the complete algorithm. Note that the number of samples ($M$) used in the reconstruction term can be different from that required for the KL estimation, which can be reduced when the model is expensive (e.g., we set $M = 1$ in the experiments of VAEs). Thus compared to normal reparameterized VI, the extra computational cost is mainly in calculating the inverse of the $n_p \times n_p$ matrix in Eq. (6). As we shall see in experiments, tens or a hundred samples are sufficient to obtain a stable KL estimate, so the added cost is not high.

---

**Algorithm 1** Kernel Implicit Variational Inference (KIVI)

---

**Require:** Observed data $\mathbf{x}$, model $p_\theta(\mathbf{x}|\mathbf{z})p(\mathbf{z})$.
**Require:** Implicit variational posterior $q_\phi(\mathbf{z}|\mathbf{x})$, $n_p$, $n_q$, $M$.
1: **repeat**
2:    Sample from the prior: $\mathbf{z}_i^p \sim p(\mathbf{z})$, $i = 1, \ldots, n_p$.
3:    Sample from the variational posterior: $\mathbf{z}_j^q \sim q_\phi(\mathbf{z}|\mathbf{x})$, $j = 1, \ldots, n_q$.
4:    Compute the density ratio $\hat{r}_{pq}$ by Eq. (5), (6) and clip $\hat{r}_{pq}$ to be positive at $\mathbf{z}^q$s.
5:    Compute $\hat{\text{KL}} = -\frac{1}{n_q}\sum_{j=1}^{n_q} \log \hat{r}_{pq}(\mathbf{z}_j^q)$ and $\hat{\mathcal{L}} = \frac{1}{M}\sum_{m=1}^{M} \log p(\mathbf{x}|\mathbf{z}_m^q) - \hat{\text{KL}}$.
6:    Estimate $\nabla_\phi \mathcal{L}$ with the reparameterization trick.
7:    Do gradient ascent with $\nabla_\phi \mathcal{L}$.
8:    (Optional) For parameter learning, do gradient ascent with $\nabla_\theta \mathcal{L}$.
9: **until** Convergence

---

KIVI addresses the two challenges stated in Sec. 2. First, the ratio estimates are given in closed-forms, thus not having the problem of not catching up. Second, the bias-variance trade-off of the estimation can be controlled by the regularization coefficient $\lambda$. When $\lambda$ is set smaller, the estimation is more aggressive to match the samples. When $\lambda$ is set larger, the estimated ratio function is smoother.

Choosing an appropriate $\lambda$, the variance of the gradients can be controlled, compared to the extreme ratio estimates given by discriminators when their output probabilities are near 0 or 1. Moreover, KIVI is directly applicable to both global and local latent variable models (LVMs), which is an advantage over nonparametric VI methods like particle mirror descent (Dai et al., 2015) and Stein variational gradient descent (SVGD) (Liu & Wang, 2016). For the task of training local LVMs like VAEs, we additionally use the adaptive contrast (AC) technique (Mescheder et al., 2017), whose details are summarized in Appendix D.

## 4   EXAMPLE: IMPLICIT VARIATIONAL BAYESIAN NEURAL NETWORKS

Now we present an example for using KIVI in Bayesian neural networks (BNNs), which have received increasing attention due to their ability to model uncertainty, an important factor in many tasks such as adversarial defense and reinforcement learning. However, despite that we have removed the need for a discriminator, it is still nontrivial to apply KIVI to BNNs because we need to design an implicit posterior that outputs very high-dimensional samples of latent variables (weights). Existing implicit posteriors (Mescheder et al., 2017; Song et al., 2017) based on traditional fully-connected neural networks cannot handle such a high-dimensional output space. We present *Matrix Multiplication Neural Network* (MMNN), an efficient architecture for sampling large matrices. Deploying MMNN, KIVI can easily scale up to large BNNs.

In BNNs, a prior is specified over the neural network parameters $\mathbf{W} = \{\mathbf{W}_l\}_{l=1}^{L}$, where $\mathbf{W}_l$ indicates weights in the $l$-th layer. Given an input $\mathbf{x}$, the output $y$ is modeled with

$$\mathbf{W} \sim N(\mathbf{0}, \mathbf{I}), \quad \hat{y} = f_{\mathrm{NN}}(\mathbf{x}, \mathbf{W}), \quad y \sim \mathcal{P}(\hat{y}; \theta),$$

where $\hat{y}$ is the output of the feed-forward network $f_{\mathrm{NN}}$, and $y$ is of a distribution $\mathcal{P}$ parameterized by $\hat{y}$ and $\theta$. For regression, $\mathcal{P}$ is usually a Gaussian with $\hat{y}$ as the mean. For classification, $\mathcal{P}$ is usually a discrete distribution with $\hat{y}$ as the unnormalized log probabilities.

The true posterior of $\mathbf{W}$ in BNNs is intractable. Thus we turn to VI and use a variational posterior $q$ to approximate it. Denoting the dataset with $\mathbf{X} = \{\mathbf{x}_i\}_{i=1}^{N}$, $\mathbf{Y} = \{y_i\}_{i=1}^{N}$, we have the ELBO:

$$\mathcal{L}(\mathbf{Y}, \mathbf{X}; \phi) = \mathbb{E}_{q_\phi(\mathbf{W})} \log p(\mathbf{Y}|\mathbf{X}, \mathbf{W}) - \mathrm{KL}(q_\phi(\mathbf{W}) \| p(\mathbf{W})).$$

The variational posterior is usually set to be factorized by layer: $q_\phi(\mathbf{W}) = \prod_{l=1}^{L} q_{\phi_l}(\mathbf{W}_l)$. However, previous methods used variational posteriors with a limited capacity for each $q_{\phi_l}(\mathbf{W}_l)$, including factorized Gaussian (Hernandez-Lobato & Adams, 2015), matrix variate Gaussian (Louizos & Welling, 2016; Sun et al., 2017) and normalizing flows (Louizos & Welling, 2017). Enabled to learn implicit variational posteriors, we propose to adopt a general distribution without an explicit density function, which has a form of

$$\mathbf{W}_l^0 \sim N(\mathbf{0}, \mathbf{I}), \quad \mathbf{W}_l^q = g_{\phi_l}(\mathbf{W}_l^0).$$

Here $g$ is a transformation parameterized by $\phi_l$, and $\mathbf{W}_l^q$ are treated as samples from $q$. The key challenge is that $\mathbf{W}_l$ are very high dimensional for moderate size neural networks. Thus, we often cannot use a fully connected neural network (MLP) as $g$. Inspired by low-rank matrix factorization (Koren et al., 2009), we propose a new kind of network called *Matrix Multiplication Neural Network* (MMNN) to serve as $g$, as shown in Alg. 2. In each layer of an MMNN, an input matrix $\mathbf{X}_i$ ($M_{in} \times N_{in}$) is left multiplied with a pa-

---

**Algorithm 2** MMNN

**Require:** Input matrix $\mathbf{X}_0$
**Require:** Network parameters $\{\mathbf{A}_i^l, \mathbf{B}_i^l, \mathbf{A}_i^r, \mathbf{B}_i^r\}_{i=1}^{L}$
 1: **for** $i = 1, \ldots, L$ **do**
 2:     Left multiplication: $\mathbf{X}_i = \mathbf{A}_i^l \mathbf{X}_{i-1} + \mathbf{B}_i^l$
 3:     Right multiplication: $\mathbf{X}_i = \mathbf{X}_i \mathbf{A}_i^r + \mathbf{B}_i^r$
 4:     **if** $i \leq L - 1$ **then**
 5:         $\mathbf{X}_i = \mathrm{Relu}(\mathbf{X}_i)$
 6:     **end if**
 7: **end for**
 8: Output $\mathbf{X}_L$

---

rameter matrix $\mathbf{A}_i^l$ ($M_{out} \times M_{in}$) and is added a bias matrix $\mathbf{B}_i^l$ ($M_{out} \times N_{in}$), then it is right multiplied with a parameter matrix $\mathbf{A}_i^r$ ($N_{in} \times N_{out}$) and is added a bias matrix $\mathbf{B}_i^r$ ($M_{out} \times N_{out}$). Finally it is passed through a nonlinear activation (e.g., ReLU). We call such a layer as an $M_{out} \times N_{out}$ Matrix Multiplication layer.

To model the implicit posterior of $\mathbf{W}_l$, we only need to randomly sample a matrix $\mathbf{W}_l^0$ of smaller size, and propagate it through the MMNN to get the output variational samples ($\mathbf{W}_l^q$):

$$\mathbf{W}_l^0 \sim N(\mathbf{0}, \mathbf{I}), \quad \mathbf{W}_l^q = \text{MMNN}_{\phi_l}(\mathbf{W}_l^0).$$

When modeling a matrix, MMNN has significant computational advantages over MLPs, due to its low-rank property. For example, to model an $M \times N$ weight matrix, consider a single-layer MMNN with the input matrix $\mathbf{W}_0$ of size $M_0 \times N_0$, the parameters needed are $\mathbf{A}_1^l, \mathbf{B}_1^l, \mathbf{A}_1^r, \mathbf{B}_1^r$ and they are in total of size $MM_0 + MN_0 + N_0N + MN$, while if a single fully-connected layer is used, the parameter size is $M_0N_0MN$, which is much larger. Thus, we can use an MMNN as $g$ in the variational posterior for normal-size neural networks. In tasks with very small networks, we still use an MLP as $g$.

## 5 RELATED WORK

Our work closely relates to the works on implicit generative models (IGMs, generative models that define implicit distributions) and density ratio estimation (DRE). IGMs have drawn much attention due to the popularity of GANs. General learning algorithms of implicit models have been surveyed in Mohamed & Lakshminarayanan (2016), where DRE plays a central role. The connection between DRE and GANs is also discussed in Uehara et al. (2016). For a comprehensive review of DRE, we refer the readers to the survey (Hido et al., 2011). We also refer the readers to many other works by Sugiyama and his collaborators on DRE, such as KLIEP (Sugiyama et al., 2008), LSIF (Kanamori et al., 2009), and DRE based on Bregman divergence (Sugiyama et al., 2012).

Our work also builds upon the recent VI methods, including stochastic approximation by mini-batches (Hoffman et al., 2013), direct gradient optimization of variational lower bounds (Paisley et al., 2012; Mnih & Gregor, 2014), and the reparameterization trick for training continuous LVMs (Kingma & Welling, 2013). Following the success of learning with IGMs, implicit distributions are applied to VI. Many of them are based on discriminators, which can be divided into two categories: prior-contrastive (PC) and joint-contrastive (JC) (Huszár, 2017). In PC methods discriminators distinguish between samples from the prior and those from the variational posterior, while in JC methods they distinguish between the model joint distribution and the joint distribution composed of the data distribution and the variational posterior. Concurrent with Huszár (2017), Mescheder et al. (2017) proposed Adversarial Variational Bayes (AVB), which is an amortized version of PC methods for training local LVMs like VAEs. Prior to Huszár (2017), similar ideas with JC methods have been proposed in ALI (Dumoulin et al., 2016) and Bi-GAN (Donahue et al., 2016). Nonparametric VI methods such as PMD (Dai et al., 2015) and SVGD (Liu & Wang, 2016) that adapt a set of particles towards the true posterior are also closely related to implicit VI. They share the similar advantage of flexible approximations. More recently, the amortized version of SVGD has been developed (Liu & Feng, 2016) and the same idea has been applied to MCMC (Li et al., 2017). It was further shown in Li & Turner (2017) that the core identity in SVGD (Stein's identity) could also be employed to approximate the gradients of implicit distributions.

## 6 EXPERIMENTS

We present empirical results on both synthetic and real datasets to demonstrate the benefits of KIVI. All implementations are based on ZhuSuan (Shi et al., 2017).

### 6.1 TOY 1-D GAUSSIAN MIXTURES

We firstly conduct a toy experiment to approximate a 1-D Gaussian mixture distribution with VI. The Gaussian mixture distribution has two equally distributed unit-variance components

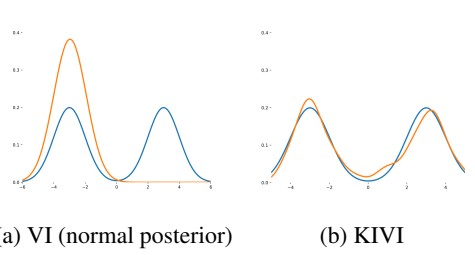

(a) VI (normal posterior)      (b) KIVI

Figure 1: Fitting Gaussian Mixture distribution

whose means are -3 and 3. We compare KIVI with VI using a single Gaussian posterior (Fig. 1). The variational distribution used by KIVI generates samples by propagating a standard normal distribution through a two-layer MLP with 10 hidden units in each layer and one output unit. As shown, the Gaussian posterior converges to a single mode. In contrast, KIVI can accurately approximate the two

Table 1: Average test set RMSE, predictive log-likelihood for the regression datasets.

| Dataset | Avg. Test RMSE | | | Avg. Test LL | | |
|---|---|---|---|---|---|---|
| | SVGD | Dropout | KIVI | SVGD | Dropout | KIVI |
| Boston | 2.957±0.099 | 2.97±0.19 | **2.798±0.173** | -2.504±0.029 | **-2.46±0.06** | -2.527±0.102 |
| Concrete | 5.324±0.104 | 5.23±0.12 | **4.702±0.116** | -3.082±0.018 | **-3.04±0.02** | -3.054±0.043 |
| Energy | 1.374±0.045 | 1.66±0.04 | **0.467±0.015** | -1.767±0.024 | -1.99±0.02 | **-1.298±0.005** |
| Kin8nm | 0.090±0.001 | 0.10±0.00 | **0.075±0.001** | 0.984±0.008 | 0.95±0.01 | **1.162±0.008** |
| Naval | 0.004±0.000 | 0.01±0.00 | **0.001±0.000** | 4.089±0.012 | 3.80±0.01 | **5.501±0.121** |
| Combined | 4.033±0.033 | 4.02±0.04 | **3.976±0.037** | -2.815±0.008 | -2.80±0.01 | **-2.794±0.009** |
| Protein | 4.606±0.013 | 4.36±0.01 | **4.255±0.019** | -2.947±0.003 | -2.89±0.00 | **-2.868±0.005** |
| Wine | **0.609±0.010** | 0.62±0.01 | 0.629±0.008 | **-0.925±0.014** | -0.93±0.01 | -0.958±0.015 |
| Yacht | 0.864±0.052 | 1.11±0.09 | **0.737±0.068** | **-1.225±0.042** | -1.55±0.03 | -2.123±0.010 |
| Year | **8.684±NA** | 8.849±NA | 8.950±NA | **-3.580±NA** | -3.588±NA | -3.615±NA |

modes with an expressive variational posterior. We defer another toy experiment on 2-D Bayesian logistic regression to Appendix F.1, where the importance of the reverse ratio trick is illustrated.

## 6.2 BAYESIAN NEURAL NETWORKS (BNNS)

As stated in Sec. 4, the latent variables in BNNs are global to all data points and are usually very high-dimensional, for which a flexible variational family is essential. We compare KIVI with state-of-the-art VI methods by doing regression and classification on standard benchmarks.

### 6.2.1 REGRESSION

To quantitatively measure the predictive ability of BNNs with KIVI as the inference method, we use standard multivariate regression benchmarks from recent works (Table 1), such as probabilistic backpropagation (PBP) (Hernandez-Lobato & Adams, 2015). We compare with state-of-the-art methods: the Bayesian interpretation of dropout (Gal & Ghahramani, 2016) and stein variational gradient descent (SVGD) (Liu & Wang, 2016) [1]. Following the setup in PBP, we use BNNs with one 50-unit hidden layer except in the two large datasets, i.e., *Protein Structure* and *Year Predication MSD*, where 100 units are used. We randomly select 90% of the whole dataset for training and leave the rest for testing. We also put a Gamma prior on the precision of the observation noise to adaptively learn it (see Appendix E). For all datasets, we set $n_p = n_q = M = 100, \lambda = 0.001$ and set the batch size to 100 and the learning rate to 0.001. The model is trained for 3000 epochs for the small datasets with less than 1000 data points, and 500 epochs for the others. We report the mean errors and standard deviations averaged over 20 runs, except 5 runs for *Protein Structure* and 1 run for *Year Predication MSD*. As networks used in these tasks are of a small scale, we use MLPs with one hidden layer in the implicit variational posterior (see Appendix G.2.1 for details).

Table 1 shows the results with the best ones marked in bold. Results of SVGD and dropout are cited from their papers, which have the same setting as ours. We can see that KIVI consistently outperforms SVGD and dropout on both RMSE and test-LL for most datasets. Especially on RMSE, KIVI has significant improvements over them except on *Wine* and *Year Predication MSD*. It suggests that KIVI enables the implicit variational posterior to capture the predictive uncertainty in network parameters, which is hard to be fully described by a mixture of two delta distributions (dropout) and a fixed set of particles (SVGD). We emphasize that although the nonparametric nature of SVGD has also made the approximation more flexible, it uses the same set of particles throughout the inference procedure, while each iteration of KIVI generates a new set of particles. Thus the implicit posterior learned by KIVI is smoothed by the parametric model. Recently, normalizing flows have shown good performance on BNNs (Louizos & Welling, 2017). So we also experiment with directly applying normalizing flows to this task. The results are reported in Appendix F.2.

| Method | # Hidden | # Weights | Test err. |
|---|---|---|---|
| SGD (Simard et al., 2003) | 800 | 1.3m | 1.6% |
| Dropout (Srivastava et al., 2014) | | | $\approx 1.3\%$ |
| Dropconnect (Wan et al., 2013) | 800 | 1.3m | **1.2%**$^\star$ |
| Bayes B. (Blundell et al., 2015), | 400 | 500k | 1.82% |
| with Gaussian posterior | 800 | 1.3m | 1.99% |
| | 1200 | 2.4m | 2.04% |
| Bayes B. (Blundell et al., 2015), | 400 | 500k | 1.36%$^\star$ |
| with scale mixture prior | 800 | 1.3m | 1.34%$^\star$ |
| | 1200 | 2.4m | 1.32%$^\star$ |
| KIVI | 400 | 500k | **1.29%** |
| | 800 | 1.3m | **1.22%** |
| | 1200 | 2.4m | **1.27%** |

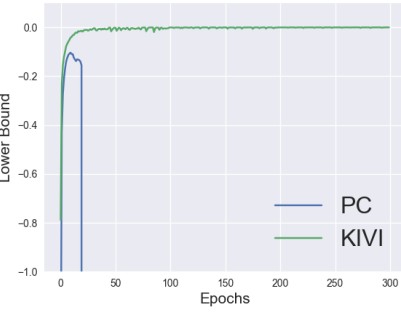

Figure 2: Results for MNIST classification. The left table shows the test error rates. $\star$ indicates results that are not directly comparable to ours: Wan et al. (2013) used an ensemble of 5 networks, and the second part of Blundell et al. (2015) changed the prior to a scale mixture. The plot on the right shows training lower bound in MNIST classification with prior-contrastive and KIVI.

### 6.2.2 CLASSIFICATION

For classification, we present the results on MNIST, which consists of 60,000 training images and 10,000 test images of handwriting digits. Compared to the above datasets in regression, MNIST has a much higher feature dimension, introducing millions of parameters even in moderate-size networks, which brings big challenges for BNNs. As a standard benchmark, the performance on MNIST can be improved by many other techniques, such as convolution, generative pre-training, data augmentation, etc. To ensure a fair comparison, we follow the settings from Bayes-By-Backprop (Blundell et al., 2015) and focus on improving the performance of ordinary MLPs without using any of these techniques. The network structures used are three MLPs, all with two ReLU hidden layers, and the layer sizes are 400, 800 and 1200, respectively. For KIVI, we used MMNNs with two hidden matrix multiplication layers in the implicit posterior (see Appendix G.2.2 for details). We set $n_p = n_q = M = 10, \lambda = 0.001$, and train for 300 epochs with the batch size as 100. The initial learning rate is 0.001 and is annealed every 100 epochs by ratio 0.5. We used the last 10,000 samples of the training set as the validation set for model selection.

Fig. 2 summarizes the results. We can see that KIVI achieves better accuracy compared to plain SGD (Simard et al., 2003), dropout (Srivastava et al., 2014) and Bayes-By-Backprop (Blundell et al., 2015) on all three types of MLPs. KIVI even performs better than Bayes-By-Backprop with a changed prior (scale mixture), which makes the model itself more flexible than ours. When the layer size is 800, our result is comparable to that of an ensemble of 5 networks with dropconnect (Wan et al., 2013), which demonstrates that the implicit posterior has been learned to account for most model uncertainty.

We also conduct experiments with the prior-contrastive (PC) method (the counterpart of AVB for global latent variables, see Sec. 2). The key challenge for applying PC here is that the posterior samples are extremely high-dimensional, and if fed into discriminators like neural networks, they will cause unaffordable computation cost. To get around this, we use a logistic regression as the discriminator in PC. The experiment settings of PC are reported in Appendix G.2.2. The training lower bounds of the two methods are plotted in Fig. 2. We can see that in the beginning they increase at the same pace, then PC fails to converge with lower bound explosion while KIVI improves consistently. The explosion is mainly because the input to the discriminator is of hundreds of thousands of dimensions, and plain logistic regression cannot produce reliable density ratio estimates. We also experiment with PC for layer size 800 and 1200. They both fail to converge in the end.

### 6.3 VARIATIONAL AUTOENCODERS

As stated in Sec. 3.2, KIVI is applicable to both global and local LVMs, and the latter can be learned using an amortized scheme (i.e., use $q_\phi(\mathbf{z}|\mathbf{x})$ instead of $q_\phi(\mathbf{z})$). Here we present an application on

---

[1]Note that VMG (Louizos & Welling, 2016) and PBP-MV (Sun et al., 2017) used adaptive weight priors, which are different from the common setting of standard normal priors; the former also additionally used variational dropout, thus their results are not directly comparable to those of the works discussed here as well as ours.

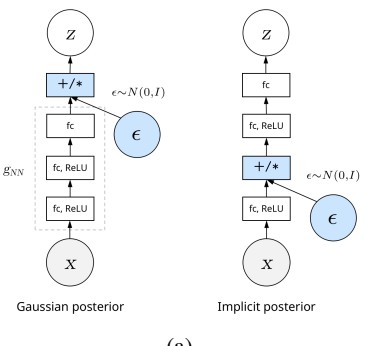
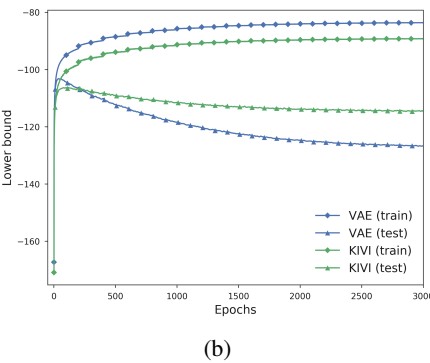

(a)                                   (b)

Figure 3: Variational Autoencoders: (a) Gaussian posterior vs. implicit posterior, where fc denotes a fully-connected layer. They are used by the plain VAE and KIVI, respectively; (b) Training and evaluation curves of the lower bounds on statically binarized MNIST.

training variational autoencoders (VAE) with implicit posteriors to demonstrate this. The generative process of VAEs proceeds as

$$\mathbf{z} \sim N(\mathbf{0}, \mathbf{I}), \quad \mathbf{x} \sim \mathcal{P}(f_{\text{NN}}(\mathbf{z})),$$

where $\mathbf{z}$ are latent features, $\mathbf{x}$ are observations, and $\mathcal{P}$ is the output distribution for modeling the observations, which takes the outputs of the neural network ($f_{\text{NN}}(\mathbf{z})$) as parameters. We conduct experiments on two widely used datasets for generative modeling: binarized MNIST and CelebA (Liu et al., 2015). $\mathcal{P}$ is a Bernoulli distribution for binarized MNIST, while a normal distribution for CelebA. For VAE the variational lower bound has the same form as Eq. (1), except that $q_\phi(\mathbf{z})$ is replaced by $q_\phi(\mathbf{z}|\mathbf{x})$. The original VAE parameterizes $q_\phi(\mathbf{z}|\mathbf{x})$ as

$$\mathbf{z} = \boldsymbol{\mu}_\phi(\mathbf{x}) + \epsilon \cdot \boldsymbol{\sigma}_\phi(\mathbf{x}), \quad \boldsymbol{\mu}_\phi(\mathbf{x}), \boldsymbol{\sigma}_\phi(\mathbf{x}) = g_{\text{NN}}(\mathbf{x}; \phi), \quad \epsilon \sim N(\mathbf{0}, \mathbf{I}),$$

where $g_{\text{NN}}$ is a neural network that outputs the parameters of the normal distribution. In the MNIST case, $g_{\text{NN}}$ is an MLP with two hidden layers, which is illustrated in Fig. 3a (left). To form an implicit posterior, a direct choice is to move the stochastic noise from the output layer to the penultimate hidden layer, as illustrated by Fig. 3a (right).

Before applying KIVI, a crucial question is what we expect to get by using implicit posteriors for training VAEs. One target could be that we may get tighter lower bounds of the data log-likelihood (LL) because the algorithm searches in a larger variational family for the optimal lower bound. This suggests, however in a very weak way, that doing optimization in the larger space will lead to better test LL, given the optimization always arrives at local optima. Previously Adversarial Variational Bayes (AVB) has shown some results on MNIST, by comparing the test LL of the plain VAE and the VAE trained by AVB, using golden truths estimated by annealed importance sampling (AIS) (Wu et al., 2016). However, for the results reported in AVB, the model architectures used by the plain VAE and AVB are very different, which leads to concerns about which part of the change contributes to the improved likelihoods.

Here we adopt another setting to better demonstrate the gain from implicit posteriors. We observe that the key improvement of implicit posteriors is that objectives with them average over a much broader range of posterior configurations. This effect not only contributes to a larger search space that contains tighter lower bound values, but also makes the VAE model better prevent overfitting. To verify that KIVI keeps this property, we conduct experiments on a statically binarized MNIST dataset and use models with no prior knowledge of the problem (MLPs instead of convnets), which is a typical setting that leads to overfitting. The latent dimension of the VAE model is 8, and $f_{\text{NN}}(\mathbf{z})$ is an MLP with two hidden layers of size 500. The parameters for KIVI are $n_p = n_q = 100, M = 1$. More details can be found in Appendix G.3. Fig 3b shows the training and testing curves of VAEs with or without KIVI. It can be seen that the lower bound gap between the training and testing curves of the plain VAE are much larger than that of KIVI, which indicates that the VAE trained by KIVI is less prone to overfitting. After 3k epochs we evaluate the test LL on 2048 test images using AIS and get **-97.3** for the plain VAE, while **-94.8** for the VAE trained by KIVI.

To demonstrate that KIVI scales to larger models, we trained a VAE with the same network structure used by DCGAN (Radford et al., 2015) on CelebA using KIVI. The latent dimension in this case is

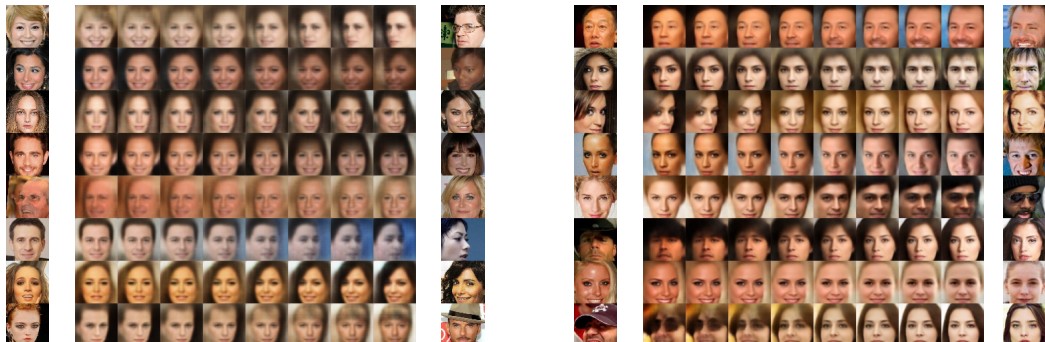

Figure 4: Interpolation experiments for CelebA: AVB (left); KIVI (right).

32. The implicit posterior is constructed in a way similar to the one shown in Fig. 3a, with the bottom hidden layers symmetric to the decoder network (See Appendix G.3 for details). To visually check the latent space learned by KIVI, we show the reconstruction results of linearly interpolating between the latent $\mathbf{z}$ vectors of two real images after 25 epochs (Fig. 4), when the model has converged. Compared to the latent space learned by AVB after the same epochs (we use the public code from AVB and set the same decoder structure), we find ours are smoother, and the interpolated images are much sharper. More interpolation results through the training process are presented and compared in Appendix F.5. We also use the learned model to generate 10,000 images and evaluate the sample quality using Fréchet Inception Distance (FID) (Heusel et al., 2017). The FID scores achieved at epoch 25 by AVB and KIVI are 160 and 41, respectively (smaller is better). In fact many efforts are required to make AVB successfully train a model for CelebA to produce results shown in the figure. As reported in Mescheder et al. (2017), the log prior $\log p(\mathbf{z})$ is explicitly added to the discriminator ($T(\mathbf{x}, \mathbf{z})$), while KIVI does not need much tuning: there is no need to carefully design a discriminator, and the only two hyper-parameters (i.e., $\lambda$ and the clipping value) both have clear meanings.

## 7 CONCLUSIONS

We present an implicit VI method named Kernel Implicit Variational Inference (KIVI), which provides a principled way of tuning bias-variance tradeoff and makes implicit VI computationally feasible for models with high-dimensional latent variables. We successfully applied this approach to Bayesian neural networks and achieved superior performance on both regression and classification tasks. We also demonstrate that KIVI can be applied to learn local latent variable models like VAEs. Future work may include applying this method to larger-scale networks and improving the kernel estimator further.

### ACKNOWLEDGMENTS

The work was supported by the National Natural Science Foundation of China (NSFC) Projects (Nos. 61620106010, 61621136008), Beijing Natural Science Foundation No. L172037, Tsinghua Tiangong Institute for Intelligent Computing, the NVIDIA NVAIL Program and a Project from Siemens.

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

# A    PROOF OF PROPOSITION 1

*Proof.* Let $V$ denote the span of the representers of the sample points:

$$V = \text{span}(\{k(\mathbf{z}_i, \cdot) : i = 1, \ldots, n_p + n_q\} = \left\{ \sum_{i=1}^{n_p} \alpha_i k(\mathbf{z}_i^p, \cdot) + \sum_{j=1}^{n_q} \beta_j k(\mathbf{z}_j^q, \cdot) \right\},$$

where we label $\mathbf{z}_{1:n_p}^p$ as $\mathbf{z}_{1:n_p}$ and $\mathbf{z}_{1:n_q}^q$ as $\mathbf{z}_{n_p+1:n_p+n_q}$. Define the orthogonal complement $V_\perp$ to be

$$V_\perp = \{f \in \mathcal{H} : \langle f, g \rangle = 0, \forall g \in V\}.$$

Because $\hat{r} \in \mathcal{H}$, it can be decomposed into two parts:

$$\hat{r} = \hat{r}_\| + \hat{r}_\perp,$$

where $\hat{r}_\| \in V$, and $\hat{r}_\perp \in V_\perp$. Note that the squared loss is not changed by the orthogonal component $\hat{r}_\perp$:

$$\hat{r}(\mathbf{z}_i) = (\hat{r}_\| + \hat{r}_\perp)(\mathbf{z}_i) = \hat{r}_\|(\mathbf{z}_i) + \langle \hat{r}_\perp, k(\mathbf{z}_i, \cdot) \rangle = \hat{r}_\|(\mathbf{z}_i).$$

Meanwhile the regularization term can be decomposed into $\|\hat{r}_\|\|_{\mathcal{H}}^2 + \|\hat{r}_\perp\|_{\mathcal{H}}^2$. So the optimal solution will have $\hat{r}_\perp^* = 0$, which indicates that $\hat{r}^* \in V$. □

# B    DERIVATION OF THE OPTIMAL DENSITY RATIO FUNCTION

Substituting Eq. (5) into Eq. (4) and removing the constant, we get

$$\mathcal{L}(\boldsymbol{\alpha}, \boldsymbol{\beta}) = \frac{1}{2n_p} \left( \|\mathbf{K}_p \boldsymbol{\alpha}\|_2^2 + \|\mathbf{K}_{pq} \boldsymbol{\beta}\|_2^2 + 2\boldsymbol{\alpha}^\top \mathbf{K}_p \mathbf{K}_{pq} \boldsymbol{\beta} \right) - \frac{1}{n_q} \left( \boldsymbol{\alpha}^\top \mathbf{K}_{pq} \mathbf{1} + \boldsymbol{\beta}^\top \mathbf{K}_q \mathbf{1} \right)$$
$$+ \frac{\lambda}{2} (\boldsymbol{\alpha}^\top \mathbf{K}_p \boldsymbol{\alpha} + \boldsymbol{\beta}^\top \mathbf{K}_q \boldsymbol{\beta} + 2\boldsymbol{\alpha}^\top \mathbf{K}_{pq} \boldsymbol{\beta}),$$

where $(\mathbf{K}_p)_{i,j} = k(\mathbf{z}_i^p, \mathbf{z}_j^p)$, $(\mathbf{K}_{pq})_{i,j} = k(\mathbf{z}_i^p, \mathbf{z}_j^q)$, and $(\mathbf{K}_q)_{i,j} = k(\mathbf{z}_i^q, \mathbf{z}_j^q)$. Taking derivatives w.r.t. $\boldsymbol{\alpha}$ and $\boldsymbol{\beta}$ and setting them to zeros:

$$\frac{\partial \mathcal{L}}{\partial \boldsymbol{\alpha}} = \frac{1}{n_p} \mathbf{K}_p (\mathbf{K}_p \boldsymbol{\alpha} + \mathbf{K}_{pq} \boldsymbol{\beta}) - \frac{1}{n_q} \mathbf{K}_{pq} \mathbf{1} + \lambda \mathbf{K}_p \boldsymbol{\alpha} + \lambda \mathbf{K}_{pq} \boldsymbol{\beta} = \mathbf{0},$$

$$\frac{\partial \mathcal{L}}{\partial \boldsymbol{\beta}} = \frac{1}{n_p} \mathbf{K}_{pq}^\top (\mathbf{K}_{pq} \boldsymbol{\beta} + \mathbf{K}_p \boldsymbol{\alpha}) - \frac{1}{n_q} \mathbf{K}_q \mathbf{1} + \lambda \mathbf{K}_q \boldsymbol{\beta} + \lambda \mathbf{K}_{pq}^\top \boldsymbol{\alpha} = \mathbf{0}.$$

Rearranging the terms, we have

$$\left( \frac{1}{n_p} \mathbf{K}_p \mathbf{K}_{pq} + \lambda \mathbf{K}_{pq} \right) \boldsymbol{\beta} + \mathbf{K}_p \left( \frac{1}{n_p} \mathbf{K}_p + \lambda \mathbf{I} \right) \boldsymbol{\alpha} = \frac{1}{n_q} \mathbf{K}_{pq} \mathbf{1}, \tag{8}$$

$$\left( \frac{1}{n_p} \mathbf{K}_{pq}^\top \mathbf{K}_{pq} + \lambda \mathbf{K}_q \right) \boldsymbol{\beta} + \mathbf{K}_{pq}^\top \left( \frac{1}{n_p} \mathbf{K}_p + \lambda \mathbf{I} \right) \boldsymbol{\alpha} = \frac{1}{n_q} \mathbf{K}_q \mathbf{1}. \tag{9}$$

Left multiplying Eq. (8) with $\mathbf{K}_{pq}^\top \mathbf{K}_p^{-1}$ and then subtracting Eq. (9) yields

$$\boldsymbol{\beta} = \frac{1}{\lambda n_q} \mathbf{1}.$$

Substituting the optimal $\boldsymbol{\beta}$ into Eq. (8), we get

$$\boldsymbol{\alpha} = -\frac{1}{\lambda n_p n_q} (\frac{1}{n_p} \mathbf{K}_p + \lambda \mathbf{I})^{-1} \mathbf{K}_{pq} \mathbf{1}.$$

Note that although $\mathbf{K}_p^{-1}$ is used in the derivation, the forms of the solution do not require $\mathbf{K}_p$ to be invertible. In fact, if $\mathbf{K}_p$ has zero eigenvalues (not invertible), the objective $\mathcal{L}(\boldsymbol{\alpha}, \boldsymbol{\beta})$ is not bounded below since one can always choose an $\boldsymbol{\alpha}$ in the null space of $\mathbf{K}_p$ to decrease it. In other words, the form of the solution is well defined in any condition and is optimal when the objective is well defined.

## C  GRADIENTS OF THE KL TERM

$$\nabla_\phi \text{KL}(q_\phi || p) = \nabla_\phi \text{E}_{q_\phi} \log \frac{q_\phi}{p}$$

$$= \int \left[ \nabla_\phi q_\phi(z) \log \frac{q_\phi(z)}{p(z)} + q_\phi(z) \nabla_\phi \log \frac{q_\phi(z)}{p(z)} \right] \text{d}z$$

$$= \nabla_\phi \text{E}_{q_\phi} \log \frac{q}{p} + \int \nabla_\phi q_\phi(z) \text{d}z$$

$$= \nabla_\phi \text{E}_{q_\phi} \log \frac{q}{p}.$$

The formula above shows that a good density ratio estimator gives accurate gradient estimation of $\phi$.

## D  KIVI WITH ADAPTIVE CONTRAST

Although KIVI gives rather good estimation of the density ratio, the estimation accuracy still degrades with larger discrepancy between $p$ and $q$. The problem is very critical for local latent variable models like VAEs because the same variational model is required to infer posteriors of all local latent variables. In order to mitigate that, AVB (Mescheder et al., 2017) adopted a technique called Adaptive Contrast (AC), which can easily be integrated with KIVI. AC introduces an auxiliary tractable distribution $r_\alpha(\mathbf{z}|\mathbf{x})$ that resembles $q$. With $r_\alpha(\mathbf{z}|\mathbf{x})$, the ELBO can be rewritten as

$$\mathcal{L}(\mathbf{x}; \phi) = \text{E}_{q_\phi}(- \log r_\alpha(\mathbf{z}|\mathbf{x}) + \log p(\mathbf{x}, \mathbf{z})) - \text{KL}(q_\phi(\mathbf{z}|\mathbf{x})||r_\alpha(\mathbf{z}|\mathbf{x})).$$

Gradients of the first term w.r.t. $\phi$ can be easily computed using Monte Carlo, and gradients of the second term can be estimated using KIVI. Adaptive contrast gives better estimates if $r_\alpha(\mathbf{z}|\mathbf{x})$ approximates $q$ well. Because $r$ is required to have a tractable density, a commonly used adaptive distribution is a Gaussian distribution whose mean $\mu_r$ and standard derivation $\sigma_r$ match with $q$. In practice $\mu_r$ and $\sigma_r$ are estimated from the samples of $q$. According to the invariance of KL divergence under reparameterization, we have

$$\text{KL}(q_\phi(\mathbf{z}|\mathbf{x})||r_\alpha(\mathbf{z}|\mathbf{x})) = \text{KL}(\hat{q}_\phi(\hat{\mathbf{z}}|\mathbf{x})||\hat{r}_0(\hat{\mathbf{z}})),$$

where $\hat{q}_\phi(\mathbf{z}|\mathbf{x})$ denotes the distribution of $\hat{\mathbf{z}} = \frac{\mathbf{z} - \mu_r}{\sigma_r}$ and $\hat{r}_0(\hat{\mathbf{z}})$ denotes the standard normal distribution. Under this reparameterization, we only need to estimate the density ratio between two distributions with zero means and unit variances.

## E  LOWER BOUND WITH GAMMA-PRIOR PRECISION

In the multivariate regression task, the output is sampled from a normal distribution with $\hat{y}(x, \mathbf{W})$ as the mean and a parameter as the variance. The variance controls the likelihood of the model, therefore, choosing an appropriate variance is essential. We place a Gamma prior $\text{Gamma}(6, 6)$ on its reciprocal (i.e., the precision of the normal distribution). The variational posterior we used is $q(\mathbf{W}, \lambda) = q(\mathbf{W})q(\lambda), q(\lambda) \sim \text{Gamma}(\alpha, \beta)$. Then the ELBO can be calculated as

$$\mathcal{L} = \mathbb{E}_{q(\mathbf{W})} \mathbb{E}_{q(\lambda)} \log p(y|\mathbf{x}, \mathbf{W}, \lambda) - \text{KL}(q(\mathbf{W})||p(\mathbf{W})) - \text{KL}(q(\lambda)||p(\lambda))$$

$$= \mathbb{E}_{q(\mathbf{W})} \mathbb{E}_{q(\lambda)} \log \mathcal{N}(y | \hat{y}(\mathbf{x}, \mathbf{W}), \frac{1}{\lambda}) - \text{KL}(q(\mathbf{W})||p(\mathbf{W})) - \text{KL}(q(\lambda)||p(\lambda))$$

$$= \frac{1}{2} \mathbb{E}_{q(\mathbf{W})} \mathbb{E}_{q(\lambda)} \left[ \log \lambda - \lambda(y - \hat{y}(\mathbf{x}, \mathbf{W}))^2 - \log 2\pi \right] - \text{KL}(q(\mathbf{W})||p(\mathbf{W}))$$
$$- \text{KL}(q(\lambda)||p(\lambda))$$

$$= \frac{1}{2} \mathbb{E}_{q(\mathbf{W})} \left[ \psi(\alpha) - \log \beta - \frac{\alpha}{\beta}(y - \hat{y}(\mathbf{x}, \mathbf{W}))^2 - \log 2\pi \right] - \text{KL}(q(\mathbf{W})||p(\mathbf{W}))$$
$$- \text{KL}(q(\lambda)||p(\lambda)),$$

where $\psi(x)$ is the digamma function and $\text{KL}(q(\lambda)||p(\lambda))$ can be calculated in closed-form.

## F  ADDITIONAL EXPERIMENT RESULTS

### F.1  2-D BAYESIAN LOGISTIC REGRESSION

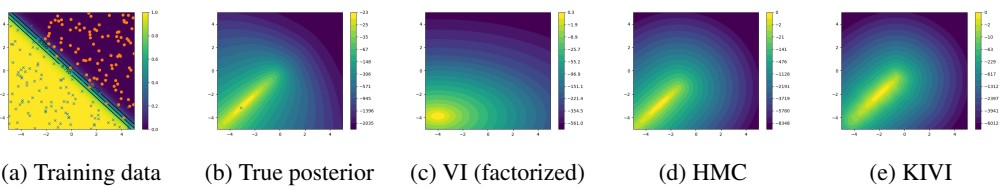

(a) Training data     (b) True posterior     (c) VI (factorized)     (d) HMC     (e) KIVI

Figure 5: 2-D Bayesian logistic regression

We also conduct experiments on a 2-D Bayesian logistic regression example, which has an intractable posterior. The model is

$$\mathbf{w} \sim N(\mathbf{0}, \mathbf{I}), \quad y_i \sim \text{Bernoulli}(\sigma(\mathbf{w}^\top \mathbf{x}_i)), \quad i = 1, \dots, N,$$

where $\mathbf{w}, \mathbf{x}_i \in \mathbb{R}^2$; $\sigma$ is the sigmoid function. $N = 200$ data points ($\{(x_i, y_i)\}_{i=1}^N$) are generated from the true model as the training data (Fig. 5a). The unnormalized true posterior is plotted in Fig. 5b. As a baseline, we first run VI with a factorized normal distribution. The result is shown in Fig. 5c. It can be clearly seen that the factorized normal can capture the position and the scale of the true posterior but cannot fit well to the shape due to its independence across dimensions.

We then apply KIVI. The implicit posterior we use is a simple stochastic neural network (see Appendix G.1). To demonstrate how good the result is, we also run Hamiltonian Monte Carlo (HMC) to get posterior samples. The results are plotted in Fig. 5d and 5e. We can see that the implicit posterior is learned to capture the strong correlation between the two dimensions and can produce posterior samples that have a similar shape with the samples drawn by HMC.

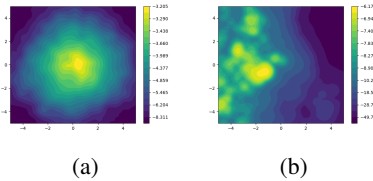

(a)          (b)

Figure 6: Variational distributions produced by only optimizing $\text{KL}(q(\mathbf{w})\|p(\mathbf{w}))$: (a) With the reverse ratio trick; (b) Without the reverse ratio trick.

We also use this experiment to illustrate the importance of the reverse ratio trick. See Figure 6 for the implicit variational distribution learned by only optimizing $\text{KL}(q_\phi(\mathbf{w})\|p(\mathbf{w}))$. In this way we expect the learned $q$ to be close to the prior $N(\mathbf{0}, \mathbf{I})$. The result produced by using the trick is compared to the result without it. We can see that the latter fails to work well.

### F.2  COMPARISON WITH MORE METHODS

In this section we present comparisons with two more methods towards flexible posteriors, namely, normalizing flow (Rezende & Mohamed, 2015) and KSD variational inference (KSD VI) (Liu & Feng, 2016). We also include the results by HMC as the ground truth.

**Normalizing flow**  The basic idea of normalizing flow has been introduced in Section 1. Specifically, given a random variable $\mathbf{z} \in \mathbb{R}^d$ following a simple distribution $q(\mathbf{z})$, and an invertible mapping $T : \mathbb{R}^d \to \mathbb{R}^d$ so that $\mathbf{z}' = T(\mathbf{z})$, the probability density of the transformed variable $\mathbf{z}'$ can be calculated as

$$q_T(\mathbf{z}') = q(\mathbf{z}) \left| \det\left(\frac{\partial T(\mathbf{z})}{\partial \mathbf{z}}\right) \right|^{-1}. \tag{10}$$

Thus we can construct complex distributions by composing invertible mappings ($T$), while keeping the probability density of the result distribution tractable by successively applying Eq. (10). For

Table 2: Test RMSE, log-likelihood for the regression datasets. Factorized and NF represent VI with factorized normal posteriors and normalizing flow, respectively.

| RMSE | Factorized | NF | KSD VI | KIVI | HMC |
|---|---|---|---|---|---|
| boston | 3.42±0.19 | 3.43±0.19 | 4.10±0.25 | 2.80±0.17 | **2.20±0.05** |
| concrete | 6.00±0.10 | 6.04±0.10 | 12.49±0.19 | 4.70±0.12 | **4.27±0.13** |
| energy | 2.42±0.06 | 2.48±0.09 | 5.54±0.14 | **0.47±0.02** | **0.47±0.01** |
| kin8nm | 0.09±0.00 | 0.09±0.00 | 0.26±0.00 | 0.08±0.00 | **0.07±0.00** |
| naval | 0.01±0.00 | 0.01±0.00 | 0.01±0.01 | **0.00±0.00** | **0.00±0.00** |
| **LL** | **Factorized** | **NF** | **KSD VI** | **KIVI** | **HMC** |
| boston | -2.66±0.04 | -2.66±0.04 | -3.32±0.01 | -2.53±0.10 | **-2.29±0.01** |
| concrete | -3.22±0.06 | -3.24±0.06 | -4.13±0.01 | -3.05±0.04 | **-2.81±0.02** |
| energy | -2.34±0.02 | -2.36±0.03 | -3.61±0.00 | **-1.30±0.01** | -1.43±0.01 |
| kin8nm | 0.96±0.01 | 1.01±0.01 | -0.18±0.01 | 1.16±0.01 | **1.22±0.01** |
| naval | 4.00±0.11 | 4.04±0.12 | 3.28±0.13 | 5.50±0.12 | **7.31±0.00** |

example, Rezende & Mohamed (2015) proposed the planar normalizing flow:

$$T(\mathbf{z}) = \mathbf{z} + \mathbf{u}h(\mathbf{w}^\top \mathbf{z} + b),$$

where $\mathbf{u}, \mathbf{w} \in \mathbb{R}^d, b \in \mathbb{R}$ are free parameters, and $h$ is a smooth element-wise nonlinearty, chosen as Tanh in the following experiments. A simple variational posterior can be made more expressive when this parametric transformation is employed.

**KSD VI** KSD variational inference (Liu & Feng, 2016) is a method that directly minimizes the kernelized Stein discrepancy (KSD) between the true posterior ($p$) and the variational posterior ($q$):

$$
\begin{aligned}
\mathbb{S}(q, p) &= \max_{f \in \mathcal{H}^d} \left( \mathbb{E}_q[s_p(\mathbf{z})^\top f(\mathbf{z}) + \nabla_\mathbf{z} \cdot f(\mathbf{z})] \right) \\
&= \mathbb{E}_{\mathbf{z}, \mathbf{z}' \sim q} \left[ s_p(\mathbf{z})^\top k(\mathbf{z}, \mathbf{z}') s_p(\mathbf{z}') + s_p(\mathbf{z})^\top \nabla_{\mathbf{z}'} k(\mathbf{z}, \mathbf{z}') \right. \\
&\quad \left. + \nabla_\mathbf{z} k(\mathbf{z}, \mathbf{z}')^\top s_p(\mathbf{z}') + \nabla_\mathbf{z} \cdot \nabla_{\mathbf{z}'} k(\mathbf{z}, \mathbf{z}') \right],
\end{aligned}
\tag{11}
$$

where $\mathcal{H}^d$ is a unit ball in the vector-valued RKHS induced by the RBF kernel $k(\mathbf{z}, \mathbf{z}') = \frac{\|\mathbf{z} - \mathbf{z}'\|_2^2}{2\sigma^2}$, and $s_q(\mathbf{z}) = \nabla_\mathbf{z} \log p(\mathbf{z}|\mathbf{x})$. Note that the objective in Eq. (11) only requires samples from $q$, so KSD VI also belongs to implicit VI methods.

We conduct the regression experiments in Section 6.2.1. The results are shown in Table 2. For normalizing flow, we apply 10 planar flows on the weights to match the running time of our implicit posteriors. To see whether flows help the inference, we also present results of VI using factorized normal distributions on the weights (with their means and standard deviations optimized by the reparameterization trick). For KSD VI, we use the same implicit posteriors as those we used for KIVI in Section 6.2.1. For VI with factorized normal distributions and normalizing flow, we use 100 samples, batch size 10 except that for *kin8nm* and *naval* we use batch size 100. We set the learning rate to 0.01 and run 500 epochs for 20 times. For KSD VI, we set all the training parameters (number of samples, number of epochs, batch size, and learning rate) the same as KIVI's. We also run HMC to produce the ground truths. For HMC, we use 20 chains, 150000 iterations and set the target acceptance rate to 97% to adapt the step size. As the HMC experiment is time-consuming, we only perform 10 runs.

From Table 2 we can see that normalizing flow does not show improvements over VI with factorized normal distributions. This is probably due to optimization challenges caused by the limited form of planar flows, thus the inference procedure takes little benefit from the flexibility introduced by the flow. For KSD VI, we found it is very sensitive to the initialization of implicit posteriors. We had to set the variance of the input Gaussian noise larger so that KSD VI would not diverge at the very beginning of training. KSD VI also soon converges to unsatisfying local optima after the optimization process starts. These two findings are well explained by the conclusion that KSD is the magnitude of a functional gradient of KL divergence (Liu & Wang, 2016), then all saddle points in the original problem of optimizing the KL divergence become local optima when optimizing KSD.

### F.3 VISUALIZATION OF INFERRED POSTERIORS

In order to visually check the quality of the uncertainty estimated by KIVI, we use a visualization technique named parallel coordinates (Inselberg & Dimsdale, 1987) to plot the posterior over weights. We compare the posteriors inferred in the regression experiment on *Boston housing* (Table 2) by VI with factorized normal approximation, HMC, and KIVI.

The feature dimension of the Boston housing data is 13. And the BNN used is an MLP with a hidden layer of size 50. So the network has two weight matrices: $\mathbf{W}_0 \in \mathbb{R}^{14 \times 50}$ and $\mathbf{W}_1 \in \mathbb{R}^{51 \times 1}$ (The bias parameters are included). Since $\mathbf{W}_0$ has 700 dimensions, we only plot the posterior samples of $\mathbf{W}_1$ to avoid visual clutter. Specifically, we draw 100 samples from the posterior: $\mathbf{W}_1^{(1:100)} \sim q(\mathbf{W}_1)$, and the goal is to show these 51-dimensional samples. In parallel coordinates each sample is plotted as a polyline (see Figure 7). The vertices on the polyline represent each single dimension. Their positions on the horizontal axis are the indices of the dimension (from 0 to 50). And the vertical coordinates represent the weight values. Note that an important fact about hidden neurons in neural networks is that they are non-identifiable (any two hidden nodes can be exchanged without affecting the output distribution). This indicates that all dimensions of $\mathbf{W}_1$ are non-identifiable. Different inference algorithms may converge to different local modes caused by this symmetry. To make the visualizations comparable, we sort the dimensions of $\mathbf{W}_1$ for each algorithm by the mean value of posterior samples in each dimension.

From the visualization we could see that BNNs with factorized normal approximation have significant over-pruning problems. A large proportion of hidden nodes are turned off during the inference and the information is mainly carried by several others. These pruned weights can be identified by their low signal-to-noise ratio $\frac{|\mu|}{\sigma}$, where $\mu$ is the mean, and $\sigma$ is the standard deviation. This problem has been pointed out previously, both in VAE works (Sønderby et al., 2016; Hoffman, 2017) and BNN works (Blundell et al., 2015), and can be explained by the looseness of the variational bound when factorized approximation is used (Trippe & Turner, 2017). In contrast, the ground truth by HMC does not have the pruning problems. And there are very strong correlations captured if we observe the neighboring dimensions.[2] KIVI also does not have the pruning problems and could capture the strong correlations across the dimensions. And with the gain in accuracy, there is still a good amount of uncertainty in the implicit posterior. Despite the weight dimensions are non-identifiable, we could still see that the two VI methods arrive at biased solutions compared to the ground truth by HMC in terms of scales. Note that this is not necessarily problematic since VI is typically known to produce biased solutions.

### F.4 ACCURACY OF KL ESTIMATION

From Section 3.1 we know that the approximate gradients of the ELBO are directly related to the KL estimates. So we would like to assess the accuracy of the KL estimator. We adopt the settings from the VAE experiment on MNIST (Section 6.3). For comparison, we must also be able to compute a ground truth of the KL term. To achieve this, we use normalizing flow in $q(\mathbf{z}|\mathbf{x})$, which has a complicated but tractable density. Thus we can get a good Monte Carlo estimate of the true KL term: $\mathrm{KL}(q_\phi(\mathbf{z}|\mathbf{x}) \| p(\mathbf{z})) \simeq \frac{1}{m} \sum_{i=1}^{m} \log \frac{q_\phi(\mathbf{z}_i|\mathbf{x})}{p(\mathbf{z}_i)}, \mathbf{z}_i \sim q_\phi(\mathbf{z}|\mathbf{x})$. In Figure 8a we compare the KL term estimated using KIVI with the ground truth. Note that since we use adaptive-contrast (AC) (see Appendix D) in the VAE experiments, where the KL term is broken down into two parts, it should make more sense to look at the only part that uses the density ratio estimator (the KL divergence between the standardized $q$ and a standard normal distribution), which is plotted in Figure 8b. We can see that the KL estimates closely track the ground truth, and are more accurate as the variational approximation improves over time.

---

[2]Note that in HMC the samples plotted are drawn from a single chain. This is also because the non-identifiability of weights, since different chains with different initializations tend to converge to different local modes that cannot be identified from each other. So if we plotted all the chains, there were too much visual clutter, and to be fair, we would need to run the two VI methods with different initializations.

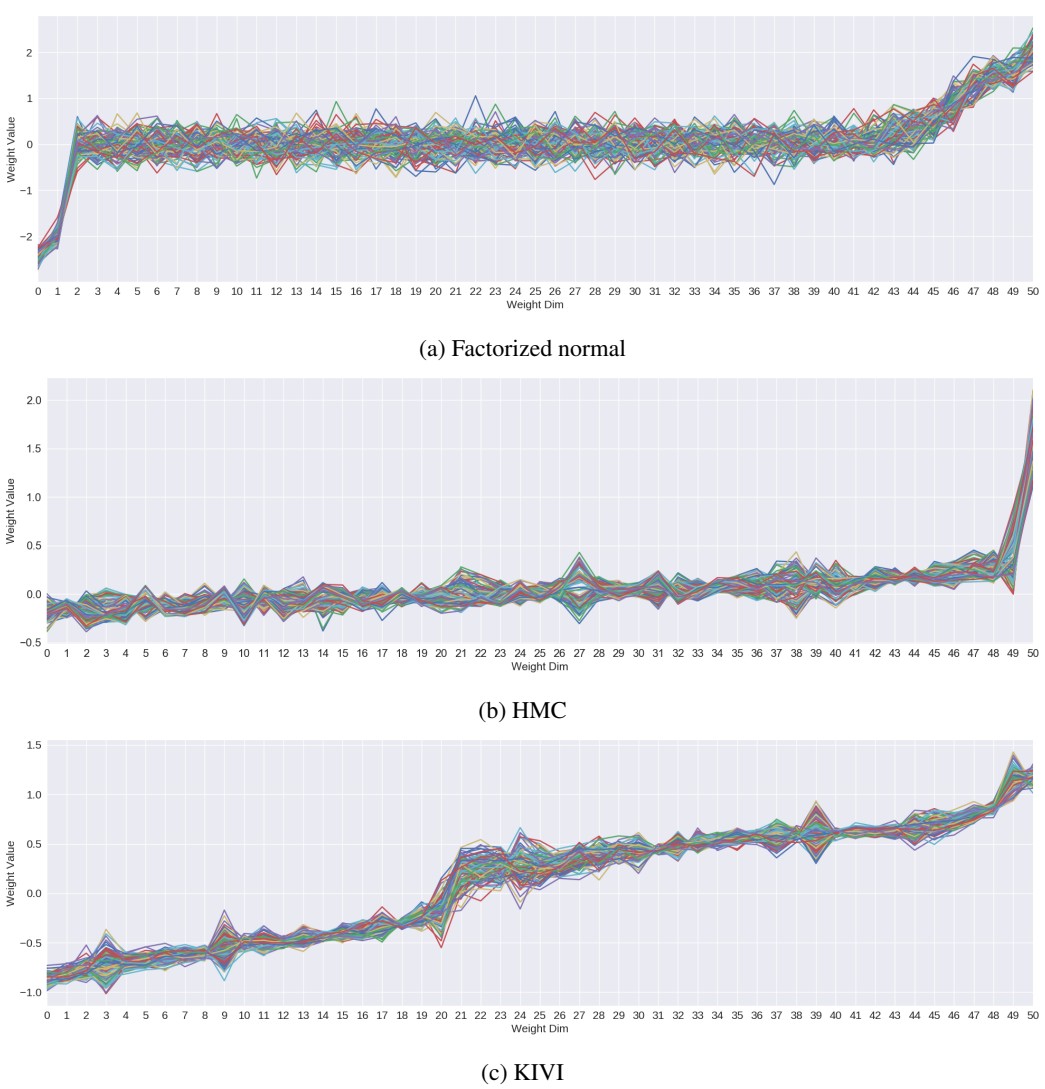

Figure 7: Visualization of learned posteriors for regression on Boston housing.

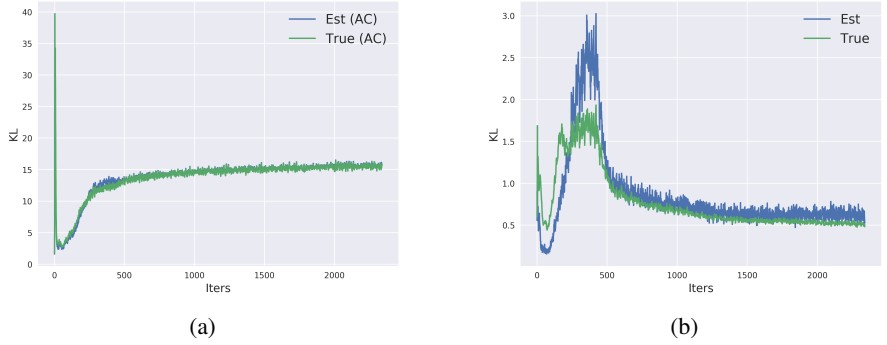

Figure 8: True KL term vs. estimated KL term for posteriors with normalizing flows.

### F.5 CHANGE OF INTERPOLATION RESULTS ON CELEBA THROUGH TRAINING

In Figures 9 to 14 we present the generated images for the interpolation experiments on CelebA through the training process. The images are generated after 1, 5, 10, 15, 20, and 25 epochs. Results on the left are produced by AVB, and on the right by KIVI. It can be clearly seen that AVB's training process is of very high variance, as we have mentioned in Section 2.

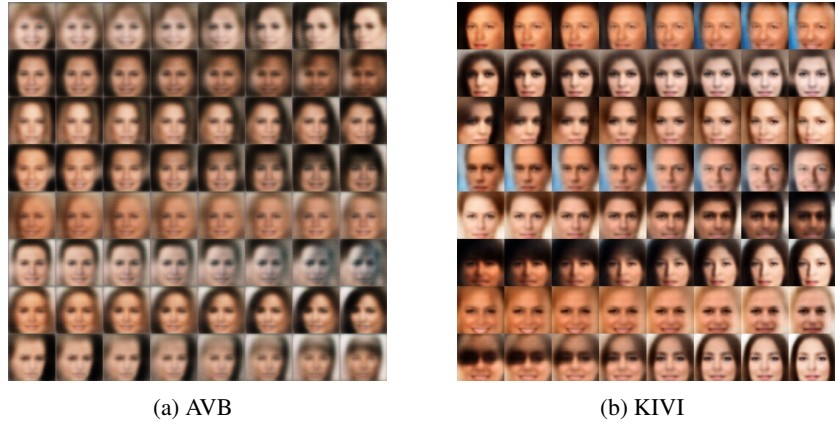

(a) AVB                          (b) KIVI

Figure 9: Epoch 1

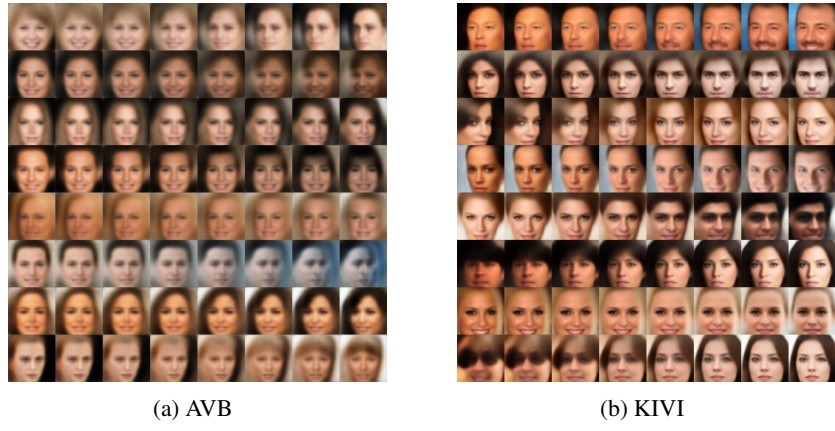

(a) AVB                          (b) KIVI

Figure 10: Epoch 5

## G   DETAILS OF EXPERIMENTS

### G.1   TOY EXPERIMENTS

**1-D Gaussian Mixture** The implicit posterior generates samples by propagating samples from a standard normal distribution through a two-layer MLP with 10 hidden units and one output unit. We set the regularization coefficient $\lambda$ to 0.003 and the density ratio clipping threshold to $10^{-8}$.

**2-D Bayesian Logistic Regression** The inputs $\mathbf{X}$ are 200 points randomly sampled from $\mathrm{U}[-5, 5] \times \mathrm{U}[-5, 5]$. The outputs $\mathbf{Y}$ are the predictions of $\mathbf{X}$ with randomly sampled weights from the prior. For HMC, we run 100 chains, 200 iterations each, and use 10 leapfrog steps. The step size is automatically adapted using dual averaging, starting from 0.001. We discard the first 100 samples generated. For factorized VI, the training is run for 100 epochs and 100 samples are used. In the training, we anneal the learning rate linearly according to $lr = \frac{100}{100+\text{epoch}-1}$. For KIVI we follow the same setting, except that we use 1k samples. The regularization coefficient $\lambda$ and the minimal density ratio are set to 0.1 and $10^{-8}$, respectively. Below we describe the sample generation process of the

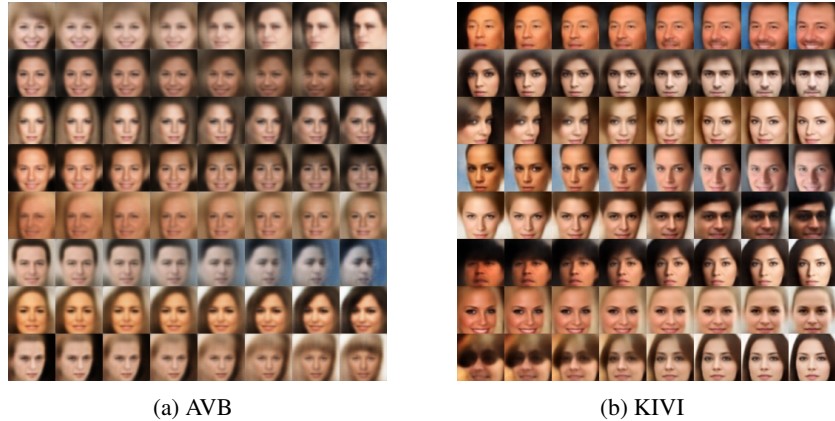

(a) AVB           (b) KIVI

Figure 11: Epoch 10

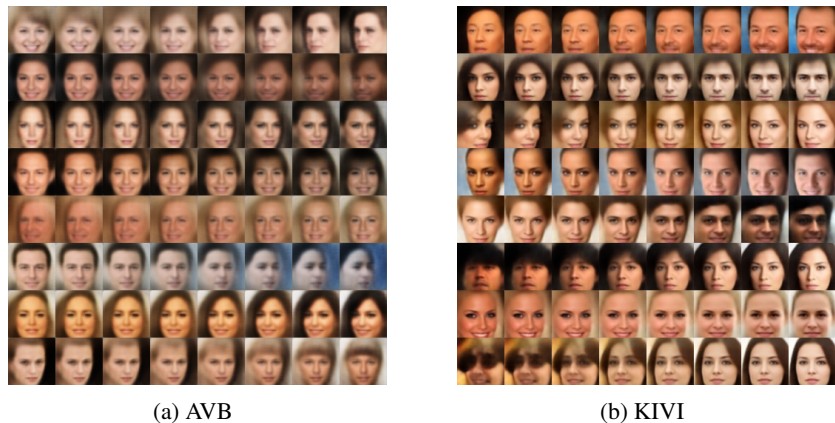

(a) AVB           (b) KIVI

Figure 12: Epoch 15

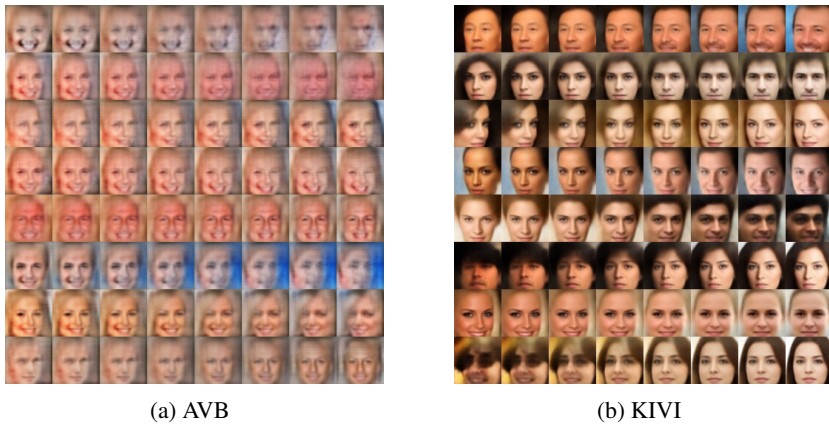

(a) AVB           (b) KIVI

Figure 13: Epoch 20

implicit variational posterior used in KIVI. First, some 2-D random normal samples are propagated through two fully-connected layers of size 20, producing $\mathbf{h}$ (we do not use activation functions for the second layer), and then $\mathbf{h}$ is added with another random normal noise with trainable variances, producing $\mathbf{z}$. Finally, we propagate $\mathbf{z}$ through a fully-connected layer of size 20 and then a linear output layer, getting the variational samples.

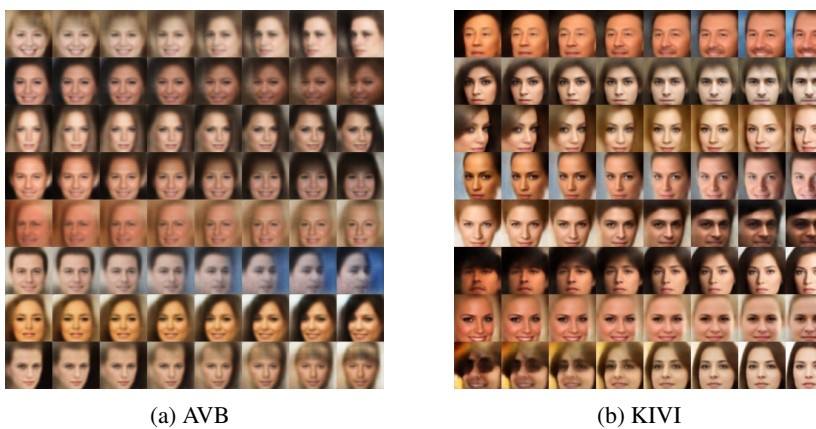

(a) AVB                           (b) KIVI

Figure 14: Epoch 25

## G.2   BAYESIAN NEURAL NETWORKS

### G.2.1   REGRESSION

As the regression datasets have small feature dimensions (all less than 15, except 90 for *Year*), using BNNs of one hidden layer (50 units) does not produce very high-dimensional weights. Therefore, we still use MLPs in the implicit variational posterior, of which the samples are generated by propagating samples from a standard normal distribution through an MLP. For all datasets, we use ReLU as the activation function. The MLP has one hidden layer except that for *Yacht* it has two hidden layers.

We list the details in Table 3, which consist of 10 datasets and 2 weight matrices each. Taking Layer 1 for *Boston* as an example, (20, 30, N1) represents that 20 random normal samples are generated and propagated through an MLP with a hidden layer of size 30 and an output layer of size N1$= 14 \times 50$. Note that N1 corresponds to the number of weights in the first layer of the BNN.

Table 3: Implicit variational posteriors for regression experiments

|         | **Boston**      | **Concrete**    | **Energy**      | **Kin8nm**          | **Naval**           |
|---------|-----------------|-----------------|-----------------|---------------------|---------------------|
| Layer 1 | (20, 30, N1)    | (30, 50, N1)    | (100, 500, N1)  | (100, 500, N1)      | (100, 500, N1)      |
| Layer 2 | (20, 30, N2)    | (30, 50, N2)    | (50, 100, N2)   | (50, 100, N2)       | (50, 100, N2)       |
|         | **Combined**    | **Protein**     | **Wine**        | **Yacht**           | **Year**            |
| Layer 1 | (100, 500, N1)  | (100, 500, N1)  | (20, 10, N1)    | (100, 800, N1, N1)  | (100, 500, N1)      |
| Layer 2 | (100, 500, N2)  | (100, 500, N2)  | (5, 20, N2)     | (50, 200, 51, N2)   | (100, 500, N2)      |

### G.2.2   CLASSIFICATION

MNIST classification needs a larger scale network than the one used in multivariate regression. Therefore, we adopt an MMNN in the implicit variational posterior. We denote the hidden-layer size of the BNN as $L$ (In our experiments $L = 400, 800$ or $1200$). Below we set $N = 500$ when $L = 400$, otherwise we set $N = 800$. In the MMNN, we use two matrix multiplication layers.

For the first-layer weights ($785 \times L$), the two hidden matrix multiplication layers are both of size $N \times N$ and are with ReLU activations. The output layer of the MMNN is of size $L \times 785$ and is with linear activations. The input matrices are random samples of size $30 \times 30$ drawn from a standard normal distribution.

For the second-layer weights ($L \times (L + 1)$), the two hidden matrix multiplication layers are both of size $N \times N$ and are with ReLU activations. The output layer of the MMNN is of size $L \times (L + 1)$ and is with linear activations. The input matrices are random samples of size $30 \times 30$ drawn from a standard normal distribution.

For the third-layer weights ($10 \times (L + 1)$), the two hidden matrix multiplication layers are both of size $30 \times N$ and are with ReLU activations. The output layer of the MMNN is of size $10 \times (L + 1)$ and is with linear activations. The input matrices are random samples of size $30 \times 30$ drawn from a standard normal distribution.

We use the above variational posterior settings in both KIVI and PC. For PC, a logistic regression serves as the discriminator. We set the regularization coefficient $\lambda$ to 0.001 and the minimal density ratio to $10^{-8}$ for KIVI. Both two methods use 10 samples, and we set the batch size to 100 and the learning rate to 0.001 in training.

### G.3 VARIATIONAL AUTOENCODERS

The decoders used in the MNIST experiment are MLPs with two hidden ReLU layers. The latent dimension is 8. Each hidden layer is of size 500. The implicit posterior is also an MLP with two hidden ReLU layers, with Gaussian noises of 500 dimensions added to the first hidden layer. The noise has zero means and 500-dimensional trainable variances. Training is with the batch size 128. The learning rate is 0.001, which is annealed by a factor of 0.5 every 200 epochs. The parameters for KIVI in this case are $n_p = n_q = 100, M = 1, \lambda = 0.001$, and the clipping value is set to $10^{-8}$.

The decoders used in the CelebA experiment have exactly the same structure with the one used for $64 \times 64$ images in the DCGAN paper (Radford et al., 2015). The latent dimension is 32. The implicit variational posterior is a deep convolutional neural network with a symmetric structure to the decoder, except that the output of the last convolutional layer is flattened and is added a Gaussian noise of the same shape as the last dimension. The noise has zero means and trainable variances. The last hidden layer is fully-connected and has 500 ReLU units. For AVB we use the same decoder. For both KIVI and AVB, we use batch size 64. The other training parameters of AVB follow from its original code for CelebA. The parameters for KIVI in this case are $n_p = n_q = 100, M = 1, \lambda = 0.001$, and the clipping value is set to $10^{-8}$. The learning rate for KIVI is 0.0003.

