# OpenReview forum: "Kernel Implicit Variational Inference"
_ICLR.cc/2018/Conference — Accept (Poster)_

### Official Review · AnonReviewer2 · 2017-11-12
**Interesting and novel idea but poor writing quality**

**Rating:** 7
**Confidence:** 3

**Review:**

This paper presents Kernel Implicit Variational Inference (KIVI), a novel class of implicit variational distributions. KIVI relies on a kernel approximation to directly estimate the density ratio. Importantly, the optimal kernel approximation in KIVI has closed-form solution, which allows for faster training since it avoids gradient ascent steps that may soon get "outdated" as the optimization over the variational distribution runs. The paper presents experiments on a variety of scenarios to show the performance of KIVI.

Up to my knowledge, the idea of estimating the density ratio using kernels is novel. I found it interesting, specially since there is a closed-form solution for this estimate. The closed form solution involves a matrix inversion, but this shouldn't be an issue, as the matrix size is controlled by the number of samples, which is a parameter that the practitioner can choose. I also found interesting the implicit MMNN architecture proposed in Section 4.

The experiments seem convincing too, although I believe the paper could probably be improved by comparing with other implicit VI methods, such as [Liu & Feng], [Tran et al.], or others.

My major criticism with the paper is the quality of the writing. I found quite a few errors in every page, which significantly affects readability. I strongly encourage the authors to carefully review the entire paper and search for typos, grammatical errors, unclear sentences, etc.

Please find below some further comments broken down by section.

Section 1: In the introduction, it is unclear to me what "protect these models" means. Also, in the second paragraph, the authors talk about "often leads to biased inference". The concept to "biased inference" is unclear. Finally, the sentence "the variational posterior we get in this way does not admit a tractable likelihood" makes no sense to me; how can a posterior admit (or not admit) a likelihood?

Section 3: The first paragraph of the KIVI section is also unclear to me. In Section 3.1, it looks like the cost function L(\hat(r)) is different from the loss in Eq. 1, so it should have a different notation. In Eq. 4, I found it confusing whether L(r)=J(r). Also, it would be nice to include a brief description of why the expectation in Eq. 4 is taken w.r.t. p(z) instead of q(z), for those readers who are less familiar with [Kanamori et al.]. Finally, the motivation behind the "reverse ratio trick" was unclear to me (the trick is clear, but I didn't fully understand why it's needed).

Section 4: The first paragraph of the example can be improved with a brief discussion of why the methods of [Mescheder et al.] and [Song et al.] "are nor applicable". Also, the paragraph above Eq. 11 ("When modeling a matrix...") was unclear to me.

Section 6: In Figure 1(a), I think there must be something wrong, because it is well-known that VI tends to cover one of the modes of the posterior only due to the form of the KL divergence (in contrast to EP, which should look like the curve in the figure). Additionally, Figure 3(a) (and the explanation in the text) was unclear to me. Finally, I disagree with the discussion regarding overfitting in Figure 3(b): that plot doesn't show overfitting because it is a plot of the training loss (and overfitting occurs on test); instead it looks like an optimization issue that makes the bound decrease.


**** EDITS AFTER AUTHORS' REBUTTAL ****

I increased the rating to 7 after reading the revised version.

---

> ### Author Response · Authors · 2017-12-29
> **Thank you for the detailed comments. We apologize for the typos and unclear sentences, and we have revised the paper.**
>
> Thank you for the detailed comments. We apologize for the typos and errors. We have corrected them and revised the unclear sentences. Below, we address the individual concerns.
>
> Q1: Comparisons with other implicit VI methods, such as [Liu & Feng], [Tran et al.], or others:
> Thanks for the suggestion. In the revision, we added the comparison with (Liu & Feng, 2016) in Appendix F.2. Their approach is to directly minimize the kernel Stein discrepancy (KSD) between the variational posterior and the true posterior. Since KSD has been shown to be the magnitude of a functional gradient of KL divergence (Liu & Wang, 2016), all saddle points in the original problem of optimizing KL divergence will become local optima when optimizing KSD. In experiments we also found that KSD VI soon converges to local minima, where the performance is unsatisfying.
>
> For (Tran et al., 2017), as it investigates both implicit models and implicit inference, the technique used is the joint-contrastive method, which is beyond our scope (only meaningful to use joint-contrastive when the model is also implicit). So the comparison is infeasible since we are only focusing on implicit inference. We have compared to other discriminator-based approaches in our experiments (e.g., prior-contrastive, AVB).
>
> Q2: Detailed comments by section:
> Section 1: We revised all the unclear statements. “biased inference” means the true posterior is far from the variational family when the family only includes factorized distributions. “admit a tractable likelihood” should be “have a tractable density”.
>
> Section 3: We revised the unclear statements. In Section 3.1, we cleaned the notations and added the description of why the expectation in Eq.4 is taken w.r.t. p(z). We also revised the reverse ratio trick part. A comparison between estimation with and without the trick is added to Appendix F.1.
>
> Section 4: The implicit distributions introduced in [Mescheder et al.] and [Song et al.] are not applicable because they are based on traditional fully-connected neural networks, which cannot afford a very large output space. However, this is indeed the case of the distribution over weights in a normal-size BNN. We made it clearer in the paper. The paragraph above the original Eq. (11) has been revised.
>
> Section 6: Thanks for pointing out the error in Figure 1(a). We have corrected it. VI with normal posterior indeed converges to a single mode. For Figure 3(a), we made it clearer and added more detailed descriptions to the posterior. Figure 3(b) did show overfitting, where we have plotted both the training and the test loss. The smaller their gap is, the less the model overfits. We added more descriptions in Section 6.3.

---

> > ### Comment · AnonReviewer2 · 2018-01-02
> > **Mostly convinced with the rebuttal; thank you for revising the paper**
> >
> > Thank you for revising the paper; I think it is much clearer now. The issues in my initial review have been appropriately taken care of.
> >
> > There are still some typos here and there, and I would recommend the authors to carefully revise the paper again. Some examples are:
> >
> > Section 2:
> > . there has been some works --> there HAVE
> > . Note that the ratio approximation ... --> this sentence is unclear to me, do you mean that the gradient of the ratio approximation is zero once the approximation is accurate? Same comment in Sec 3.1, below Eq. 7.
> > . doesn't --> does not (same goes for it's, won't, etc., in other sections)
> >
> > Section 3:
> > . Why does notation change from q_phi(z) to just q(z)?
> > . Substitute Eq. (5) --> SubstitutING Eq. (5) and SETTING the derivatives ... TO zero, we ...

---

> > > ### Author Response · Authors · 2018-01-06
> > > **Thank you for reading the rebuttal**
> > >
> > > Thank you for the updated review. We answer the further questions below.
> > >
> > > Q: "Note that the ratio approximation ..." is not clear:
> > > A: This sentence has the same meaning as the one below Eq. 7, by which we mean that the true gradients of the KL term w.r.t. $\phi$ do not flow through the density ratio function, so we could replace the ratio function with its estimate who has zero gradients  w.r.t. $\phi$. We made it clearer in the paper.
> > >
> > > Q: Other typos:
> > > A: We uploaded a new revision correcting them.

---

### Official Review · AnonReviewer3 · 2017-11-25
**A good contribution to implicit approximate posterior fitting**

**Rating:** 7
**Confidence:** 4

**Review:**

Thank you for the feedback, and I think many of my concerns have been addressed.

I think the paper should be accepted.

==== original review ====

Thank you for an interesting read.

Approximate inference with implicit distribution has been a recent focus of the research since late 2016. I have seen several papers simultaneously proposing the density ratio estimation idea using GAN approach. This paper, although still doing density ratio estimation, uses kernel estimators instead and thus avoids the usage of discriminators.

Furthermore, the paper proposed a new type of implicit posterior approximation which uses intuitions from matrix factorisation. I do think that another big challenge that we need to address is the construction of good implicit approximations, which is not well studied in previous literature (although this is a very new topic). This paper provides a good start in this direction.

However several points need to be clarified and improved:
1. There are other ways to do implicit posterior inference such as amortising deterministic/stochastic dynamics, and approximating the gradient updates of VI. Please check the literature.
2. For kernel based density ratio estimation methods, you probably need to cite a bunch of Sugiyama papers besides (Kanamori et al. 2009).
3. Why do you need to introduce both regression under p and q (the reverse ratio trick)? I didn't see if you have comparisons between the two. From my perspective the reverse ratio trick version is naturally more suitable to VI.
4. Do you have any speed and numerical issues on differentiating through alpha (which requires differentiating K^{-1})?
5. For kernel methods, kernel parameters and lambda are key to performances. How did you tune them?
6. For the celebA part, can you compute some quantitative metric, e.g inception score?

---

> ### Author Response · Authors · 2017-12-29
> **Thank you for the review, and answering the questions**
>
> Thank you for the positive feedback. We address the individual questions below.
>
> Q1: Related works:
> Thanks for the suggestion. We have cited a paper on amortizing the deterministic dynamics of SVGD (Liu & Feng, 2016). In the revision, we added two more recent papers on amortized MCMC (Li et al., 2017) and gradient estimators of implicit models (Li & Turner, 2017) in Section 5. We also added more content there to highlight the contributions that Sugiyama and his collaborators has made to density ratio estimation.
>
> Q2: On the reverse ratio trick:
> In fact, we didn’t do regression under q. We only adopted the regression under p (the reverse ratio trick) in our experiments (See Algo. 1). And we have explained why the reverse ratio version is more suitable for VI in Section 3.1. In the revision, we further added a comparison between the two using the 2-D Bayesian logistic regression example in Appendix F.1, which shows that the trick is very essential for KIVI to work well.
>
> Q3: Speed and numerical issues on differentiating through alpha:
> Because K is of size n_p x n_p (n_p is the number of samples), which is usually of tens or a hundred, the cost of differentiating through K^{-1} is not high. And we used the automatic differentiation in Tensorflow. We didn’t observe any numerical issues, as long as the regularization parameter isn’t extremely small, say, less than 1e-7.
>
> Q4: Tuning parameters:
> As mentioned in Section 3.1, we selected the kernel bandwidth by the commonly used median heuristic, i.e., the kernel bandwidth is chosen as the median of pairwise distances between the samples.
>
> As for lambda, it has clear meaning, which controls the balance between bias and variance. So a good criterion would be tuning it to achieve a good trade-off between the aggressiveness of the estimate and stability of training. In the toy experiments, we tuned lambda so that optimizing only the KL term will make the posterior samples more disperse like the prior. In most other experiments, lambda is set at 0.001 which has good performance, though it could be improved by cross-validation.
>
> Q5: Quantitative evaluation for CelebA:
> Thanks for the suggestion. In fact, inception score is only suitable to natural image datasets like Cifar10 and ImageNet. Instead, we adopted a recently developed quantitative measure named Fréchet Inception Distance (FID) (Heusel et al., 2017), which improved the Inception score to use the statistics of real world samples. The scores achieved at epoch 25 by AVB and KIVI are 160 and 41 (smaller is better), respectively. We added these results in Section 6.3.

---

### Official Review · AnonReviewer1 · 2017-11-27
**Interesting idea; not clear it scales; needs experiments on quality of ratio estimation and also posterior approximation**

**Rating:** 5
**Confidence:** 4

**Review:**

Update: I read the other reviews and the authors' rebuttal. Thanks to the authors for clarifying some details. I'm still against the paper being accepted. But I don't have a strong opinion and will not argue against so if other reviewers are willing.

------

The authors propose Kernel Implicit VI, an algorithm allowing implicit distributions as the posterior approximation by employing kernel ridge regression to estimate a density ratio. Unlike current approaches with adversarial training, the authors argue this avoids the problems of noisy ratio estimation, as well as potentially high-dimensional inputs to the discriminator.  The work has interesting ideas. Unfortunately, I'm not convinced that the method overcomes these difficulties as they argue in Sec 3.2.

An obvious difficulty with kernel ridge regression in practice is that its complete inaccuracy to estimate high-dimensional density ratios.  This is especially the case given a limited number of samples from both p and q (which is the same problem as previous methods) as well as the RBF kernel. While the RBF kernel still takes the same high-dimensional inputs and does not involve learning massive sets of parameters, it also does not scale well at all for accurate estimation. This is the same problem as related approaches with Stein variational gradient descent; namely, it avoids minimax problems as in adversarial training by implicitly integrating over the discriminator function space using the kernel trick.

This flaw has rather deep implications. For example, my understanding of the implicit VI on the Bayesian neural network in Sec 4 is that it ends up as cross-entropy minimization subject to a poorly estimated KL regularizer. I'd like to see just how much entropy the implicit approximation has instead of concnetrating toward a point; or more directly, what the implicit posterior approximation looks like compared to a true posterior inferred by, say, HMC as the ground truth. This approach also faces difficulties that the naive Gaussian approximation applied to Bayesian neural nets does not: implicit approximations cannot exploit the local reparameterization trick and are therefore limited to specific architectures that does not involve sampling very large weight matrices.

The authors report variational lower bounds, which I'm not sure is really a lower bound. Namely, the bias incurred by the ratio estimation makes it difficult to compare numbers. An obvious but very illustrative experiment I'd like to see would be the accuracy of the KL estimator on problems where we can compute it tractably, or where we can Monte Carlo estimate it very well under complicated but tractable densities. I also suggest the authors perform the experiment suggested above with HMC as ground truth on a non-toy problem such as a fairly large Bayesian neural net.

---

> ### Author Response · Authors · 2017-12-29
> **Thank you for the insightful comments and we have included further experiments to investigate the questions raised.**
>
> Thank you for the insightful comments and we have included further experiments to investigate the questions raised. We have revised the paper to include the analysis.
>
> Q1: Inaccuracy of kernel regression in high dimensions & not convinced that KIVI overcomes the difficulties:
>
> First, we have to emphasize that implicit VI is surely a much harder problem than VI with a common variational posterior (e.g., Gaussian), due to the lack of a tractable density for variational posterior q. Given the limited number of samples from q per iteration, if no additional knowledge is available, almost all implicit VI methods as well as nonparametric methods (e.g., SVGD) suffer to some degree in high dimensions, as agreed by the reviewer. However, as we extensively investigated in experiments, though not fully addressed all the challenges, KIVI can outperform existing strong competitors to get state-of-the-art performance. We think this is a valuable contribution to variational inference.
>
> Below, we further clarify our contributions. We have also revised the two challenges in Section 2 and the statements of contributions in the paper to make them clearer.
>
> 1) For the noisy estimation, we focused on the variance introduced in discriminator-based methods. In fact, existing discriminator-based methods have been identified to have high variance (noisy), i.e., samples from the two distributions are easily discriminated, which indicates overfitting (Mescheder et al., 2017). This phenomenon is like the case when you push $\lambda$ in KIVI to 0. We are not claiming high accuracy for estimation in high-dimensional spaces (In fact no implicit VI method can claim that with limited samples per iteration, as explained above). One main contribution of KIVI is to provide an explicit trade-off between bias and variance, since there was no principled way of doing so in discriminator-based methods. As a result, our algorithm can be rather stable (see Fig.2, right). It’s true that bringing down the variance requires to pay some bias in the gradient in general. However, as empirically shown in the experiments and also in the investigation of the learned posteriors (see the answer to Q3 below), we found that we still gain over previous VI methods, both in terms of accuracy and also the quality of uncertainty, which is highly non-trivial.
>
> 2) For high-dimensional latent variables, the argument mainly focused on computation issues. The other main contribution of KIVI is to make implicit VI computationally FEASIBLE for models like moderate-sized BNNs. In the classification case, the weights are of tens of thousands of dimensions and can hardly be fed into neural nets, which renders discriminator-based approaches infeasible.
>
> Finally, we’d like to add a point that KIVI opens up the door for improving implicit VI methods. The view of kernel regression at least brings two possible directions: One is pointed out by the reviewer, the RBF kernel could be replaced by other kernels that are more suitable to the model here. The other is to improve the regression problem to utilize the geometry of the distribution. And the latter is actually an ongoing work of us.
>
> Q2: Accuracy of the KL estimator on problems where we can compute it tractably:
> Thanks for the suggestion. We added Appendix F.4 to compare the true KL term with the estimated KL term. We used normalizing flow there as the “complicated but tractable densities”. We can see that the KL estimates closely track the ground truth, and are more accurate as the variational approximation improves over time.
>
> Q3: Quality of posterior approximation & comparison to HMC:
> We added Appendix F.3 to visualize the posterior approximation by KIVI and compare with HMC and the VI with naive Gaussian posteriors. The quantitative results and settings of HMC are described in Appendix F.2. The main conclusion is that the VI with naive Gaussian posteriors leads to over-pruning problems. KIVI doesn’t have the problem, and retains a good amount of uncertainty compared to HMC.
>
> Q4: Cannot use local reparameterization trick:
> This is a valid point. But the problem exists as long as we want to go beyond tractable variational posteriors (e.g., Gaussian). The results by naive Gaussian posteriors have been shown above, which has significant over-pruning problems. New difficulty introduced shouldn’t be the reason that we stick to the naive Gaussian approximation.
>
> Q5: Bias of lower bounds:
> There are two places where we report lower bounds. In Figure 2 (right) the lower bounds are used only to show the stability of training. In Figure 3(b) lower bounds are plotted to show the overfitting problems. We argue that though the lower bounds have bias, their relative gap (the training/test gap) should be comparable. Moreover, in this case we have also evaluated the test log likelihoods using golden truths estimated by Annealed Importance Sampling (AIS). The results by AIS confirmed the conclusion that the KIVI-trained VAE less overfits.

---

### Author Response · Authors · 2017-11-14
**Errata**

Figure 1(a) in the toy experiment is incorrectly drawn and thus misinterpreted. The correct figure should be that the Gaussian posterior covers the left mode instead of being between the two modes, since it is initialized from left (see https://drive.google.com/file/d/1nJAVH2-Fl0P6ei-ZwBI3_Z6BvFFAYk9E/view?usp=sharing). The figure of KIVI is correct and we double-checked all the others. We sincerely apologize for this error and will fix it in the revision.

---

### Author Response · Authors · 2018-01-06
**Summary of the revision**

We thank the reviewers for all the comments and questions. Here we summarize the major changes made in the revision.

* We revised the statements of motivations and contributions in Section 1-3 to make them clearer.
* We added Appendix F.4 to compare the true KL term with the estimated KL term under “complicated but tractable densities” (normalizing flows).
* Comparisons with KSD VI (Liu & Feng, 2016) and HMC are added to Appendix F.2.
* We added Appendix F.3 to visualize the posterior approximation by KIVI and compare with HMC and VI with naive Gaussian posteriors.
* We revised the reverse ratio trick part and added a comparison between estimation with and without the trick in Appendix F.1.
* The related work section is extended to include the works pointed out by AnonReviewer3.
* In Section 6.3, we added quantitative evaluation for CelebA using the recently developed Fréchet Inception Distance (FID) (Heusel et al., 2017).
* We corrected the error in Figure 1(a).
* We fixed other typos, grammar errors, and unclear sentences.

---

### Decision · Program_Chairs · 2018-01-29
**ICLR 2018 Conference Acceptance Decision**

**Decision:**

Accept (Poster)

**Comment:**

Thank you for submitting you paper to ICLR. This paper was enhanced noticeably in the rebuttal period and two of the reviewers improved their score as a result. There is a good range of experimental work on a number of different tasks. The addition of the comparison with Liu & Feng, 2016 to the appendix was sensible. Please make sure that the general conclusions drawn from this are explained in the main text and also the differences to Tran et al., 2017 (i.e. that the original model can also be implicit in this case).